DOI: 10.1038/s41467-018-06356-1 | OPEN

# Genome-wide analyses identify a role for SLC17A4 and AADAT in thyroid hormone regulation

Alexander Teumer, Layal Chaker, Stefan Groeneweg, Yong Li, Celia Di Munno, Caterina Barbieri et al.[#]

Thyroid dysfunction is an important public health problem, which affects 10% of the general population and increases the risk of cardiovascular morbidity and mortality. Many aspects of thyroid hormone regulation have only partly been elucidated, including its transport, metabolism, and genetic determinants. Here we report a large meta-analysis of genome-wide association studies for thyroid function and dysfunction, testing 8 million genetic variants in up to 72,167 individuals. One-hundred-and-nine independent genetic variants are associated with these traits. A genetic risk score, calculated to assess their combined effects on clinical end points, shows significant associations with increased risk of both overt (Graves' disease) and subclinical thyroid disease, as well as clinical complications. By functional follow-up on selected signals, we identify a novel thyroid hormone transporter (SLC17A4) and a metabolizing enzyme (AADAT). Together, these results provide new knowledge about thyroid hormone physiology and disease, opening new possibilities for therapeutic targets.

Thyroid dysfunction is a common clinical condition, affecting ~10% of the general adult population[1]. Adequate thyroid hormone levels are essential for normal growth and differentiation, regulation of energy metabolism, and physiological function of virtually all human tissues. Thyroxine (T4) is the prohormone produced by the thyroid, which is largely converted into the active hormone 3,3′,5-triiodothyronine (T3) in peripheral tissues. Circulating T4 levels are regulated by the hypothalamus–pituitary–thyroid (HPT) axis, in which pituitary thyroid-stimulating hormone (TSH) stimulates T4 production. In turn, T4 and T3 negatively regulate TSH synthesis via a negative feedback loop.

To exert their actions, T4 and T3 cross the membranes of target cells via specific transporters. Once intracellular, they are metabolized, including the conversion of T4 to T3, followed by binding of T3 to its nuclear receptor to regulate transcription of target genes. Both T4 and T3 transport and metabolism are therefore key determinants of thyroid hormone action.

In daily clinical practice, thyroid function is assessed by measuring circulating TSH and free T4 (FT4) levels, with increased TSH indicating hypothyroidism and decreased TSH indicating hyperthyroidism. FT4 levels are decreased in overt hypothyroidism, increased in overt hyperthyroidism and in the reference range in subclinical hypo and hyperthyroidism. In the last decade, it has become clear that not only overt but also subclinical hypo and hyperthyroidism are associated with several pathological conditions, such as atrial fibrillation, coronary heart disease, stroke, depression, as well as cardiovascular and overall mortality[2–7]. More recently, studies have shown that even variation in thyroid function within the normal range is associated with many of these complications[4,8–10]. Despite the physiological significance of thyroid hormones, as well as the prevalence and clinical importance of thyroid dysfunction, many key players in the regulation of thyroid hormone bioavailability and action, including its transport and metabolism, still need to be elucidated.

Genome-wide association studies (GWAS) performed so far have revealed genetic variants in about 30 loci robustly associated with thyroid function[11–13]. However, these variants explain only <9% of the heritability of TSH and FT4 variation[14], while in total, it has been estimated at 65 and 39–80% for TSH and FT4, respectively[15,16], suggesting that many loci still await discovery.

Here, we report the results of a large meta-analysis of GWAS for circulating TSH and FT4 levels, as well as for hypo and hyperthyroidism, followed by independent replication and functional studies. Results are complemented with genetic risk score (GRS) analyses, gene expression, co-localization analyses, and associations with various clinical phenotypes (Supplementary Figure 1) to discover new pathways underlying thyroid function and disease. We identify 109 significant independent genetic associations with these traits. The GRS shows a significant association with increased risk of both Graves' disease and subclinical thyroid disease, as well as clinical complications. Finally, we identify a novel thyroid hormone transporter and a metabolizing enzyme. Together, these results enhance our knowledge about thyroid hormone physiology and disease.

## Results

**New loci affecting thyroid hormone levels**. Our GWAS meta-analyses and replication in up to 72,167 subjects of European ancestry with TSH levels within the reference range (Supplementary Data 1) discovered 19 novel loci for circulating TSH levels and 16 novel loci for circulating FT4 levels (Tables 1 and 2, Supplementary Figures 2–5), leading to a total of 42 and 21 known and novel associated loci for these two traits. As illustrated in Fig. 1, TSH and FT4 capture distinct and complementary

genetic underpinnings of thyroid function. Some of the novel loci include genes that have been previously implicated in thyroid development (*GLIS3*), thyroid hormone action and transport (*NCOR1*, *TTR*, *SLCO1B1*), thyroid hormone metabolism (*DIO2*, *DIO3OS*), and thyroid cancer (e.g., *HES1*, *SPATA13*, *DIRC3*, *ID4*) by various candidate gene studies of monogenic diseases and animal models. Multiple independent variants were found for *PDE8B*, *DIO1*, *DIO2*, *TSHR*, and *CAPZB*.

Across the 42 TSH and 21 FT4 loci, allelic heterogeneity (i.e., independently associated single-nucleotide polymorphisms (SNPs) at the same locus) was detected at 11 and 7 loci, respectively, by using linkage structure information and summary statistics-based conditional analyses (Supplementary Tables 1 and 2). All significant associations together accounted for 33% and 21% of the genetic variance of TSH and FT4, respectively, explained by all common and low-frequency variants with a minor allele frequency (MAF) >1%.

Since TSH and FT4 regulation are inversely correlated through the HPT axis, we investigated the association of the TSH-associated loci with FT4 levels, and vice versa. As shown in Fig. 1 and in Supplementary Tables 1 and 2, we observed overlapping associations (Bonferroni-corrected threshold $p < 8.2 \times 10^{-4}$) at various loci (TSH: *FGF7*, *PDE8B*, *DET1*, *ITPK1*, *VEGFA*, *GLIS3*, *NFIA*, and *MBIP*, and FT4: *FOXE1* and *GLIS3*), although only *GLIS3* showed genome-wide significance for both traits. All alleles associated with higher TSH were associated with lower FT4, with the exception of *MBIP* and *FOXE1*.

Hypo and hyperthyroidism are more prevalent in women than in men. However, sex-stratified GWAS meta-analyses for TSH and FT4 did not show any significant gene-by-sex interaction in our samples (Supplementary Figure 6, Supplementary Tables 1 and 2).

Given the high degree of functional homology between the mouse and human genome, we selected from The International Mouse Phenotyping Consortium database[17] genes that when manipulated in mice cause abnormal thyroid physiology (i.e., hormone levels, $n = 26$) or morphology ($n = 51$), and assessed whether their human homologs contained SNPs significantly ($p < 2.5 \times 10^{-5}$ for physiology and $p < 1.9 \times 10^{-5}$ for morphology, see Methods) associated in our FT4 and TSH GWAS (Supplementary Data 2). Of these candidate genes, SNPs in *CGA* (rs6924373) and *TPO* (rs9678281) contained significant associations that did not reach genome-wide significance in our GWAS for TSH. These associations were tested for replication in 9011 independent samples and achieved genome-wide significance for TSH ($p < 5 \times 10^{-8}$) in the combined dataset (Supplementary Figure 7A). Overall, these results highlight the potential of nested candidate gene approaches in GWAS summary results and emphasize the functional conservation of genes regulating thyroid function between mice and humans.

**Relation to hypo and hyperthyroidism**. Genetic variants that determine variation in circulating TSH and FT4 levels within the reference range (i.e., the individual HPT-axis setpoint) are expected to differ from variants that underlie thyroid dysfunction (hypo or hyperthyroidism). To clarify this, we also conducted a case–control GWAS meta-analysis of increased TSH levels (i.e., hypothyroidism), including cases with TSH levels above the cohort-specific reference range ($n = 3340$) and controls with TSH levels within the reference range ($n = 49,983$). The decreased TSH level (i.e., hyperthyroidism) GWAS meta-analysis included cases with TSH below the reference range ($n = 1840$ cases) and the same controls as in the increased TSH GWAS. The distribution of sex and age groups of these subjects is provided in Supplementary Table 3. Since in both GWAS analyses, cases were defined on the

**Table 1 Novel GWAS loci associated with TSH**

| SNP | Chr:position | Locus | A1/A2 | AF1 | Effect | SE | P | $I^2$ | $P_{het}$ | N | SNP function | P hyperthyroidism | P hypothyroidism |
|---|---|---|---|---|---|---|---|---|---|---|---|---|---|
| rs6724073 | 2:218,236,786 | DIRC3 | t/c | 0.74 | 0.045 | 0.007 | 3.1E−10 | 29.4 | 0.041 | 61058 | Intron | 1.1E−01 | 6.3E−02 |
| rs28502438 | 3:149,220,109 | TM4SF4 | t/c | 0.57 | 0.035 | 0.006 | 7.3E−10 | 0.0 | 0.853 | 63299 | Intron | 7.6E−01 | 7.2E−02 |
| rs13100823 | 3:185,514,088 | IGF2BP2 | t/c | 0.30 | −0.042 | 0.006 | 4.1E−12 | 2.1 | 0.432 | 63299 | Intron | 8.5E−04 | **1.7E−04** |
| rs59381142 | 3:193,916,181 | HES1 | a/g | 0.24 | −0.054 | 0.007 | 3.6E−15 | 0.0 | 0.801 | 61059 | Unknown | 1.8E−01 | 2.4E−02 |
| rs1265091 | 6:31,108,129 | PSORS1C1 | t/c | 0.19 | 0.058 | 0.007 | 5.0E−15 | 40.4 | 0.005 | 64423 | Near gene-3 | 3.0E−01 | 3.0E−01 |
| rs56009477 | 8:23,356,964 | SLC25A37 | a/g | 0.84 | 0.050 | 0.008 | 1.1E−10 | 0.0 | 0.955 | 63299 | Unknown | 9.2E−03 | 3.5E−02 |
| rs10957494 | 8:70,365,025 | SULF1 | a/g | 0.69 | −0.036 | 0.006 | 3.6E−09 | 21.4 | 0.111 | 63299 | Unknown | 3.5E−02 | 2.5E−01 |
| rs118039499 | 8:133,771,635 | TG | a/c | 0.97 | 0.185 | 0.020 | 2.9E−21 | 28.6 | 0.042 | 66615 | Intron | **1.8E−12** | 4.0E−01 |
| rs2739067* | 8:133,951,991 | TG | a/g | 0.60 | −0.042 | 0.006 | 2.4E−11 | 0.0 | 0.540 | 54288 | Intron | 1.2E−01 | 1.3E−01 |
| rs9298749 | 9:16,214,340 | C9orf92 | a/c | 0.59 | −0.038 | 0.006 | 1.6E−10 | 10.4 | 0.280 | 63299 | Unknown | 9.0E−01 | 6.9E−03 |
| rs11255790 | 10:8,682,180 | GATA3 | t/c | 0.30 | −0.039 | 0.006 | 2.5E−10 | 0.0 | 0.738 | 63299 | Unknown | 1.3E−01 | 7.9E−01 |
| rs4933466 | 10:89,849,519 | PTEN | a/g | 0.60 | 0.037 | 0.006 | 2.2E−10 | 24.3 | 0.079 | 63299 | Unknown | 2.6E−01 | 6.6E−03 |
| rs4445669 | 11:115,045,237 | CADM1 | t/c | 0.45 | −0.039 | 0.006 | 3.6E−12 | 0.0 | 0.854 | 63299 | Untranslated-3 | 5.1E−02 | 9.4E−02 |
| rs7329958 | 13:24,782,080 | SPATA13 | t/c | 0.35 | −0.044 | 0.006 | 7.1E−14 | 0.0 | 0.913 | 63299 | Intron | 6.1E−01 | 4.5E−03 |
| rs11159482* | 14:81,490,842 | TSHR | t/c | 0.09 | 0.085 | 0.013 | 6.3E−11 | 0.0 | 0.727 | 54288 | Intron | 1.9E−01 | **1.8E−04** |
| rs59334515* | 14:81,594,143 | TSHR | t/c | 0.22 | −0.054 | 0.007 | 1.1E−13 | 25.7 | 0.080 | 54288 | Intron | 1.4E−02 | 3.1E−03 |
| rs12893151 | 14:81,619,945 | TSHR | a/c | 0.21 | −0.057 | 0.007 | 2.3E−15 | 27.4 | 0.052 | 63299 | Unknown | **3.2E−04** | **1.9E−04** |
| rs1045476 | 16:4,015,313 | ADCY9 | t/c | 0.17 | 0.047 | 0.007 | 3.2E−11 | 0.0 | 0.979 | 72167 | Untranslated-3 | 1.7E−01 | 4.6E−03 |
| rs30227 | 16:14,405,428 | MIR365A | t/c | 0.61 | −0.046 | 0.005 | 2.3E−17 | 3.0 | 0.415 | 72167 | Intron | 2.6E−02 | 1.1E−01 |
| rs77819282 | 17:44,762,589 | NSF | a/g | 0.24 | 0.043 | 0.007 | 4.3E−10 | 0.0 | 0.653 | 62192 | Intron | 5.2E−01 | 7.1E−03 |
| rs1157994 | 17:59,338,574 | BCAS3 | a/g | 0.05 | −0.083 | 0.014 | 4.0E−09 | 21.0 | 0.120 | 59243 | Intron | 1.8E−01 | 4.1E−01 |
| rs12390237 | 23:3,612,081 | PRKX | a/g | 0.62 | −0.046 | 0.007 | 1.7E−11 | 0.0 | 0.760 | 36501 | Intron | 1.0E−02 | 6.6E−01 |

The table contains the list of the index SNPs and additional independent associations of replicated TSH susceptibility loci. The values are provided for the combined discovery and replication sample, for additional independent hits (*) for the discovery stage only
Bold values of the hyper and hypothyroidism p-values indicate significance after Bonferroni correction for the 61 independent TSH-associated SNPs tested ($p < 8.2E-4$)
A1 effect allele, AF1 allele frequency of A1, SE standard error of the effect, P association p-value, $I^2$ percentage of total variation across studies that is due to heterogeneity, N sample size

**Table 2 Novel GWAS loci associated with FT4**

| SNP | Chr:position | Locus | A1/A2 | AF1 | Effect | SE | P | $I^2$ | $P_{het}$ | N | SNP function |
|---|---|---|---|---|---|---|---|---|---|---|---|
| rs4954192 | 2:135,632,98 | ACMSD | t/c | 0.43 | −0.033 | 0.006 | 9.3E−09 | 2.6 | 0.424 | 62,680 | Intron |
| rs6785807 | 3:181,718,601 | SOX2-OT | a/g | 0.15 | −0.057 | 0.009 | 6.9E−11 | 7.4 | 0.340 | 55,096 | Intron |
| rs10946313 | 6:19,381,386 | ID4 | t/c | 0.63 | 0.044 | 0.006 | 6.2E−12 | 0.0 | 0.907 | 55,096 | Unknown |
| rs9356988 | 6:25,777,481 | SLC17A4 | a/g | 0.27 | −0.052 | 0.007 | 5.7E−14 | 0.0 | 0.745 | 55,096 | Intron |
| rs137964359* | 6:26,001,742 | SLC17A4 | t/c | 0.99 | −0.200 | 0.032 | 2.1E−10 | 0.0 | 0.479 | 49,269 | Unknown |
| rs17185536 | 6:100,620,931 | LOC728012 | t/c | 0.24 | 0.071 | 0.008 | 2.7E−20 | 0.0 | 0.973 | 53,801 | Unknown |
| rs67583169 | 8:61,212,179 | CA8 | c/g | 0.86 | 0.062 | 0.009 | 7.1E−12 | 0.0 | 0.936 | 53,801 | Unknown |
| rs10119187 | 9:4,223,660 | GLIS3 | t/c | 0.81 | 0.048 | 0.008 | 8.0E−10 | 6.4 | 0.357 | 56,936 | Intron |
| rs10818937 | 9:127,015,440 | NEK6 | t/c | 0.32 | −0.039 | 0.006 | 4.9E−11 | 15.8 | 0.196 | 63,971 | Unknown |
| rs11039355 | 11:47,737,501 | FNBP4 | t/c | 0.34 | −0.039 | 0.006 | 7.9E−11 | 12.2 | 0.258 | 62,677 | Near gene-5 |
| rs4149056 | 12:21,331,549 | SLCO1B1 | t/c | 0.84 | −0.048 | 0.007 | 6.3E−11 | 0.0 | 0.636 | 67,091 | Missense |
| rs150816132* | 14:80,464,293 | DIO2 | a/g | 0.01 | −0.220 | 0.040 | 3.5E−08 | 24.1 | 0.122 | 38,640 | Unknown |
| rs978055* | 14:80,534,869 | DIO2 | a/t | 0.38 | 0.038 | 0.007 | 1.1E−08 | 10.5 | 0.296 | 49,269 | Unknown |
| rs225014 | 14:80,669,580 | DIO2 | t/c | 0.64 | 0.047 | 0.006 | 4.6E−17 | 0.0 | 0.702 | 63,971 | Missense |
| rs12323871* | 14:101,852,075 | DIO3OS | t/c | 0.82 | −0.047 | 0.008 | 1.4E−08 | 25.7 | 0.091 | 49,269 | Unknown |
| rs11626434 | 14:101,998,443 | DIO3OS | c/g | 0.36 | 0.053 | 0.007 | 1.7E−16 | 40.2 | 0.006 | 55,095 | Unknown |
| rs12907106 | 15:63,873,658 | USP3 | c/g | 0.27 | −0.039 | 0.007 | 3.7E−08 | 0.0 | 0.529 | 53,801 | Intron |
| rs8063103 | 16:12,703,395 | SNX29 | c/g | 0.85 | −0.051 | 0.009 | 7.8E−09 | 7.9 | 0.335 | 53,801 | Unknown |
| rs11078333 | 17:16,049,626 | NCOR1 | a/t | 0.51 | 0.042 | 0.006 | 2.0E−12 | 0.0 | 0.513 | 62,677 | Intron |
| rs56069042 | 18:57,914,644 | MC4R | a/g | 0.95 | 0.099 | 0.017 | 3.6E−09 | 0.0 | 0.735 | 58,197 | Unknown |

The table contains the list of the index SNPs and additional independent associations of replicated FT4 susceptibility loci. The values are provided for the combined discovery and replication sample, for additional independent hits (*) for the discovery stage only
A1 effect allele, AF1 allele frequency of A1, SE standard error of the effect, P association p-value, $I^2$ percentage of total variation across studies that is due to heterogeneity, N sample size

basis of a TSH level above or below the reference range, these groups included subjects with overt but also mild subclinical forms of hypothyroidism and hyperthyroidism, respectively.

We detected seven loci for hypothyroidism and eight loci for hyperthyroidism (Supplementary Figure 8, Supplementary Table 4). At some of the loci, the variant was significantly associated with both hypo and hyperthyroidism, with effects in opposing directions. For example, a variant at PDE10A (rs2983514) was associated with both higher risk of hypothyroidism and lower risk of hyperthyroidism. Some of the hypothyroidism loci had already previously been implicated in hypothyroidism through GWAS, including TPO, FOXE1, VAV3, and a variant in ATXN2 (rs597808) in high linkage disequilibrium (LD) with the R262W polymorphism in SH2B3[18]. However, we did not detect variants in a number of well-known autoimmune thyroid disease genes (e.g., CTLA4, HLA class I and II). This may be due to the

fact that, in these population-based cohorts, patients receiving medication for autoimmune thyroiditis were excluded. Thus, thyroid autoimmunity caused by auto-antibodies may have a different set of predisposing variants. All variants associated with hyperthyroidism have not been previously found in association with hyperthyroidism, except for FOXE1[19]. However, all of these variants were in high LD with variants associated with TSH or FT4 levels within the reference range in the current or previous GWAS[11,13]. The same holds true for variants associated with hypothyroidism, suggesting that the effects of many genetic variants on thyroid function extend beyond the physiological range, thus affecting the risk of thyroid dysfunction.

As complementary analyses to investigate whether the TSH loci are also related to autoimmune thyroid diseases, we tested all variants or their proxies for association with thyroid peroxidase antibody (TPOAb) positivity of a former GWAS[20] as an early

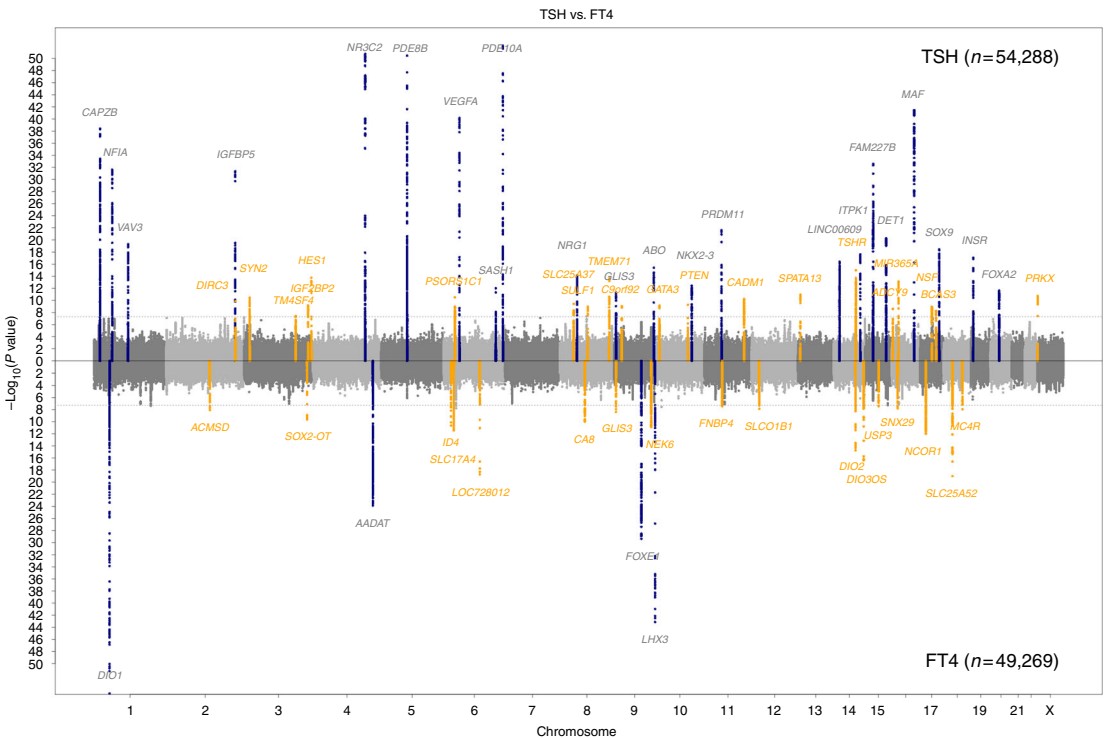

**Fig. 1** Manhattan plots for GWAS meta-analyses of thyroid function. Manhattan plots of the GWAS meta-analysis results for TSH and FT4 contrasted with each other. SNPs are plotted on the x axis according to their position on each chromosome with $-\log_{10}$(p-value) of the association test on the y axis. The upper solid horizontal line indicates the threshold for genome-wide significance, i.e., $5 \times 10^{-8}$. Genomic loci previously known to contain trait-associated variants are colored in blue, new loci in orange

marker of autoimmune hypothyroidism, as well as in patients with Graves' disease (i.e., autoimmune hyperthyroidism) from the BioBank Japan Project. For TPOAb positivity, significant associations were found for *MAF*, *SPATA13*, and *VAV3* (Supplementary Table 5). *SPATA13* and *VAV3* have previously been linked to self-reported diagnosed hypothyroidism[18,21], while no studies have investigated their potential autoimmune origin. The observed associations of these gene variants with variation in TSH levels within the normal range could therefore be due to a mild early stage of thyroid autoimmunity, instead of reflecting physiological differences in the HPT-axis setpoint.

For Graves' disease, only the psoriasis[22] *PSORS1C1* locus showed a significant association, consistent with shared genetic determinants between these two autoimmune diseases[23].

Detailed results of the mouse candidate analysis, results of pathway analyses, and look-ups for pleiotropy of the TSH, FT4, hypo and hyperthyroidism loci are described in Supplementary Note 1–3.

**Gene expression analyses**. To obtain insights into gene expression patterns and potential effector transcripts at the identified loci, we assessed whether the 94 independent index variants from the TSH, FT4, hypo and hyperthyroidism GWAS were correlated with transcript levels of nearby (*cis-*) or distant (*trans-*) genes. The results of 22 published expression quantitative trait loci (eQTL) studies were interrogated (Methods), assessing the relation between the genetic variants and gene expression patterns in a total of 127 different tissues and cell types. First, we evaluated the presence of eQTLs in at least one tissue or cell type: 38 variants showed eQTL effects (Fig. 2a and Supplementary Data 3). While many variants were associated with transcript expression in only one or few (≤8) tissues, two variants located on chromosome 17 showed ubiquitous associations with gene

expression: the FT4-associated variant rs11078333 at *NCOR1* locus and the TSH-associated variant rs199461 at the *NSF* locus. The FT4-increasing allele at rs11078333 was associated with higher expression levels of *NCOR1* in blood and brain, but also affected the expression of *ADORA2B* (increased) and *ZSWIM7* and *TTC19* (decreased) in many other tissues (including thyroid for *TTC19*). *NCOR1* is an essential nuclear co-repressor that is recruited by thyroid hormone receptors in the absence of thyroid hormone to mediate transcriptional repression. At the *NSF* locus, the TSH increasing allele at rs199461 increased expression of *KANSL1* and *LRRC37A2* and decreased expression of *WNT3* in several tissues, including thyroid. Consistent with known thyroid physiology, the majority of TSH-associated variants acted as eQTLs in thyroid tissue (Fig. 2a), with 45% of the variants being thyroid-specific eQTLs. In contrast, none of the nine FT4 eQTLs acted exclusively on the thyroid but were also associated with transcript expression changes in multiple known thyroid hormone effector organs, including liver, muscle, and adipose tissue.

Second, we used a summary-based Mendelian randomization (SMR) method coupled with testing for heterogeneity of effects (HEIDI) to assess co-localization, i.e., to investigate whether the overlap between eQTLs and GWAS hits could be attributable to the same underlying causative variant[24]. In thyroid tissue, we found evidence for co-localization with differential gene expression at 13 different GWAS loci: *PD8EB*, *PRDM11*, *MBIP* (with *RP11-116N8.1* expression), *NKX2-3*, *NSF* (with *WNT3* expression), *IGF2BP2*, *FOXA2*, *SLC25A37*, and *C9orf92* for TSH; *AADAT*, *NEK6* (with both *NEK6* and *PSMB7* expressions) for FT4; and *TPO*, *PDE8B*, and *PDE10A* for hypothyroidism (Supplementary Data 4). At these loci, our findings implicate the causal gene among the many genes present in the locus. For example, the FT4-associated variant rs6854291-influenced transcript levels of *AADAT* in thyroid while there were no effects on

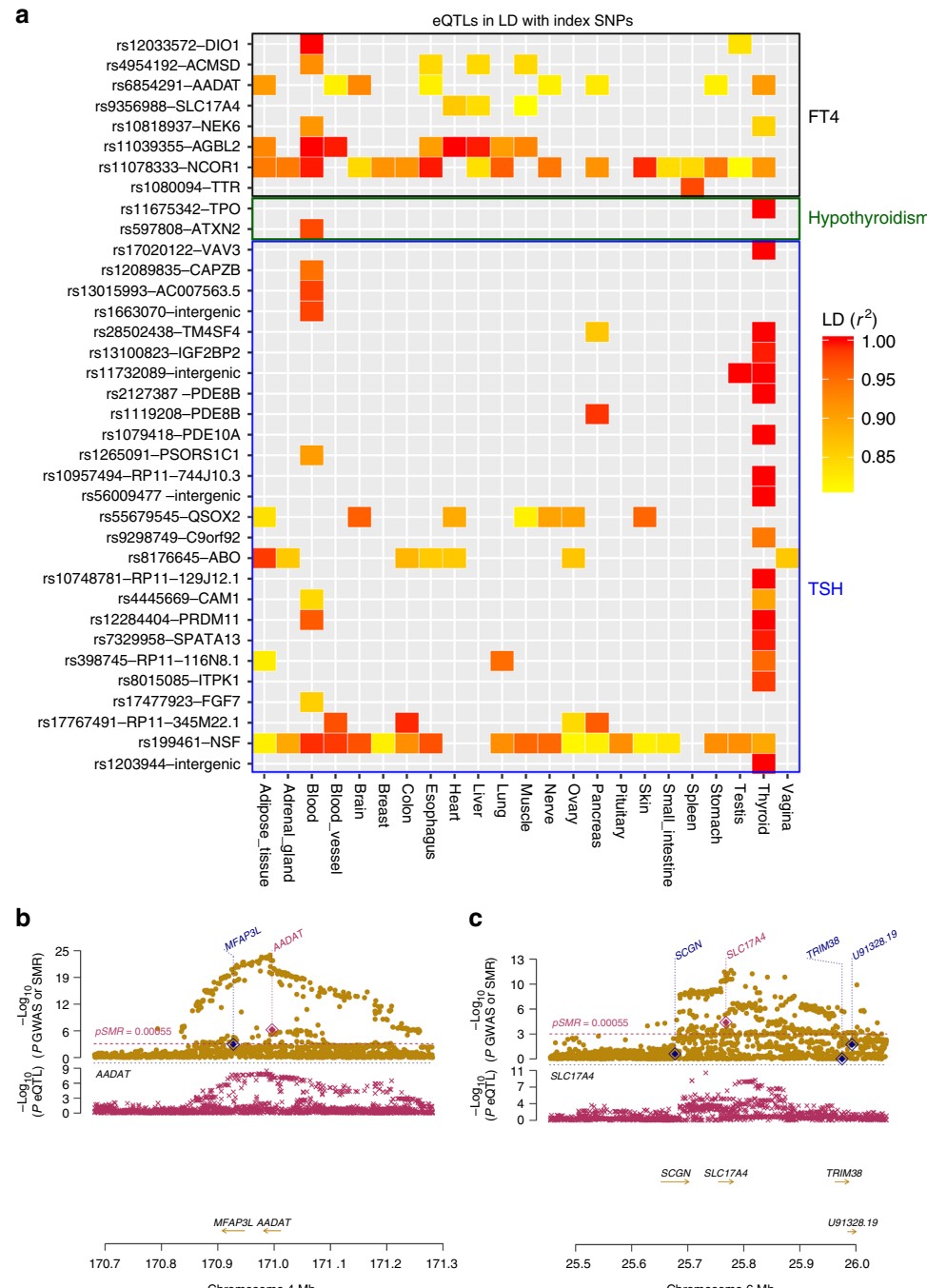

**Fig. 2** Impact on gene expression of index SNP. **a** shows tissues in which an expression QTL (eQTL) was found in LD ($r^2 > 0.8$) with FT4, hypothyroidism, and TSH index SNPs. SNPs are ordered according to trait they are associated with and then by genomic position; squares are colored according to the LD between the eQTL and the index variant, as depicted in the legend. When multiple eQTLs were detected in the same tissue, the eQTL with the highest LD is shown. **b** and **c** illustrate results of the summary-based Mendelian randomization (SMR) test for FT4 levels and expression QTLs at *AADAT* and *SLC17A4* loci, respectively. The upper box shows the regional association curve with FT4 levels, with level of significance of the SMR test (y axis) for each transcript in the locus indicated by a diamond positioned at the center of the transcript. A significant SMR test indicates an association of the transcript level of the respective genes with the trait. The lower box shows the regional association distribution with changes in expression of the highlighted transcript in pancreas. In both boxes, x axis refers to GRCh37/hg19 genomic coordinates

transcript levels of the neighboring *MFAP3L* gene, implicating *AADAT* as the causal gene underlying the FT4 association at this locus (Fig. 2b, Supplementary Data 4). We also observed that independent variants at the same locus were associated with gene expression in different tissues. For example, while the index variant rs6885099 at *PDE8B* co-localized with changes in *PDE8B*

expression in thyroid, the independent variant rs1119208 was associated with *PDE8B* expression in pancreas (Supplementary Data 4). Notably, also the variants at *AADAT* and *SLC17A4* co-localized with gene expression in pancreas (Fig. 2c), which is of interest given the close interrelations between thyroid hormone signaling, insulin regulation, and glucose metabolism[25,26].

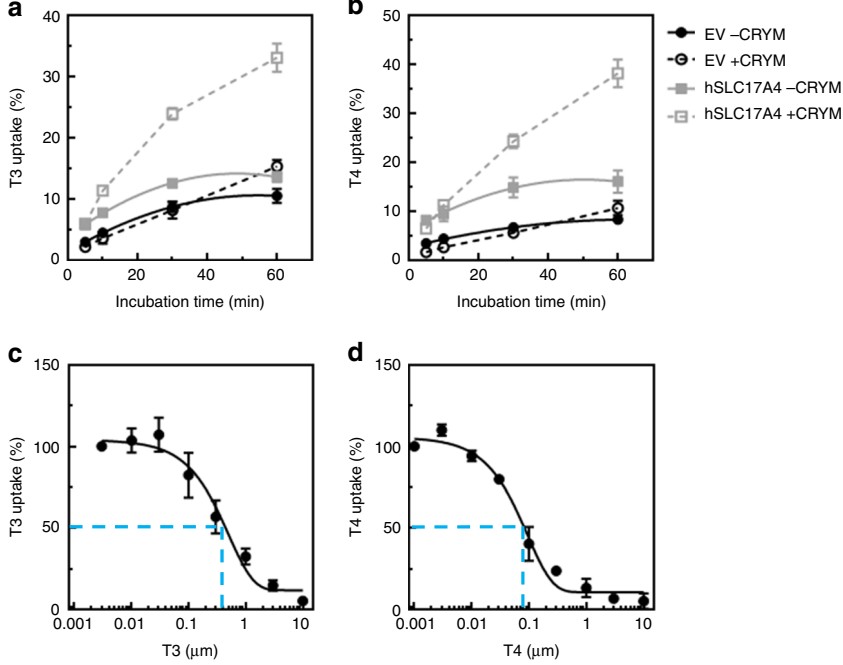

**Fig. 3** Thyroid hormone transport by hSLC17A4. Cellular T3 (**a**) and T4 (**b**) accumulation in COS-1 cells, transiently transfected with empty vector (EV), or wild-type hSLC17A4 in the absence (solid lines) or presence (dashed lines) of the intracellular thyroid hormone-binding protein CRYM, after indicated incubation times at 37 °C. All uptake levels are expressed relative to the amount of radio-labeled T3 or T4 added to the cells at the start of the incubation (1 nM (5 × 10E4 c.p.m.) [$^{125}$I]-T3 or [$^{125}$I]-T4). All results are presented as means ± SEM ($n = 4$). In the presence and absence of CRYM, T3 and T4 accumulation in hSLC17A4 transfected cells was significantly higher compared to empty-vector control cells at all time points (one-way ANOVA with a Bonferroni-corrected post hoc test, $p < 0.001$). T3 (**c**) and T4 (**d**) saturation curves in COS-1 cells transiently transfected with hSLC17A4 in the absence of CRYM. All data points are corrected for background thyroid hormone uptake in control cells and presented relatively to the amount of internalized thyroid hormone in the presence of the lowest substrate concentration (0.003 μM for T3 and 0.001 μM for T4, respectively). Apparent IC$_{50}$ values were determined by standard second order polynomial regression analyses implemented in GraphPad Prism (La Jolla, USA)

**In vitro studies**. Thyroid hormone action in target tissues is importantly determined by the amount of T3 available for receptor binding inside the cell. Therefore, the transport of thyroid hormone across the cell membrane and its metabolism inside the cell represent crucial regulatory layers in thyroid hormone signaling. Although several key players in thyroid hormone signaling have been described over the last decades, including deiodinases and several thyroid hormone transporters, many others remain to be identified. Based on their associations with circulating FT4 levels and the co-localization studies, *SLC17A4* and *AADAT* were further studied in vitro to explore a direct role in thyroid hormone signaling.

SLC17A4 is an organic anion transporter that is particularly expressed in the liver, kidney, and gastrointestinal tract[27]. We transiently over-expressed human *SLC17A4* (hSLC17A4) in COS-1 cells and observed increased cellular T3 (Fig. 3a) and T4 (Fig. 3b) accumulation compared to empty-vector transfected control cells. These effects were even stronger upon co-transfection with the intracellular thyroid hormone-binding protein mu-crystallin (CRYM) (Fig. 3a, b) and were similar in magnitude to those obtained by the monocarboxylate transporter (MCT) 8 (Supplementary Figure 9), the most specific thyroid hormone transporter identified to date. Saturation experiments in the absence of CRYM showed a dose-dependent decrease in the uptake of T3 (Fig. 3c) and T4 (Fig. 3d). The estimated IC$_{50}$ values for T3 (0.35 ± 0.13 μM, $n = 4$) and T4 (0.06 ± 0.01 μM, $n = 4$) transport by SLC17A4 are considerably lower than those of MCT8 (T3: 20.61 ± 1.26 μM and T4: 23.22 ± 1.22 μM, $n = 3$, Supplementary Figure 9), and other currently known thyroid hormone transporters[28–33], and indicate a high substrate affinity.

Together, these findings strongly indicate that *SLC17A4* encodes a high-affinity T3 and T4 transporter.

*AADAT* encodes a mitochondrial aminotransferase with broad substrate specificity, which acts on kynurenic acid and α-aminoadipate, important intermediates in tryptophan, and lysine metabolism[34,35]. The association of circulating FT4 with the *AADAT* locus suggested that AADAT may also be involved in thyroid hormone metabolism. In that case, it could facilitate the oxidative deamination of the alanine side-chain of thyroid hormone, yielding a pyruvic acid moiety[36]. Therefore, lysates of AADAT over-expressing COS-1 cells were incubated with T4 and T3 in the presence of the co-factor pyridoxal phosphate and the co-substrate α-ketoglutaric acid, and the reaction mixtures were analyzed by ultra-performance liquid chromatography (UPLC). The results demonstrated effective time- and AADAT concentration-dependent conversion of T4 and T3 to their pyruvic acid metabolites TK3 and TK4 (Fig. 4), with saturation occurring at substrate concentrations between 10 and 100 μM. Importantly, this is well below the reported Km values of AADAT for α-aminoadipate (0.9 mM) and kynurenine (4.7 mM)[34].

Given the observed effects of *SLC17A4* and *AADAT* on T3 and T4 transport and metabolism, we additionally tested the associations of the identified genetic variants in *SLC17A4* and *AADAT* with T3 levels and the T3/T4 ratio (Supplementary Table 6). *SLC17A4*-rs9356988 was associated with the T3/T4 ratio, while *AADAT*-rs6854291 was associated with both the T3/T4 ratio and T3 levels.

**Genetic TSH and FT4 risk score associations**. To assess the cumulative clinical impact of our GWAS findings, we calculated a weighted GRS for TSH and FT4 levels, which included all

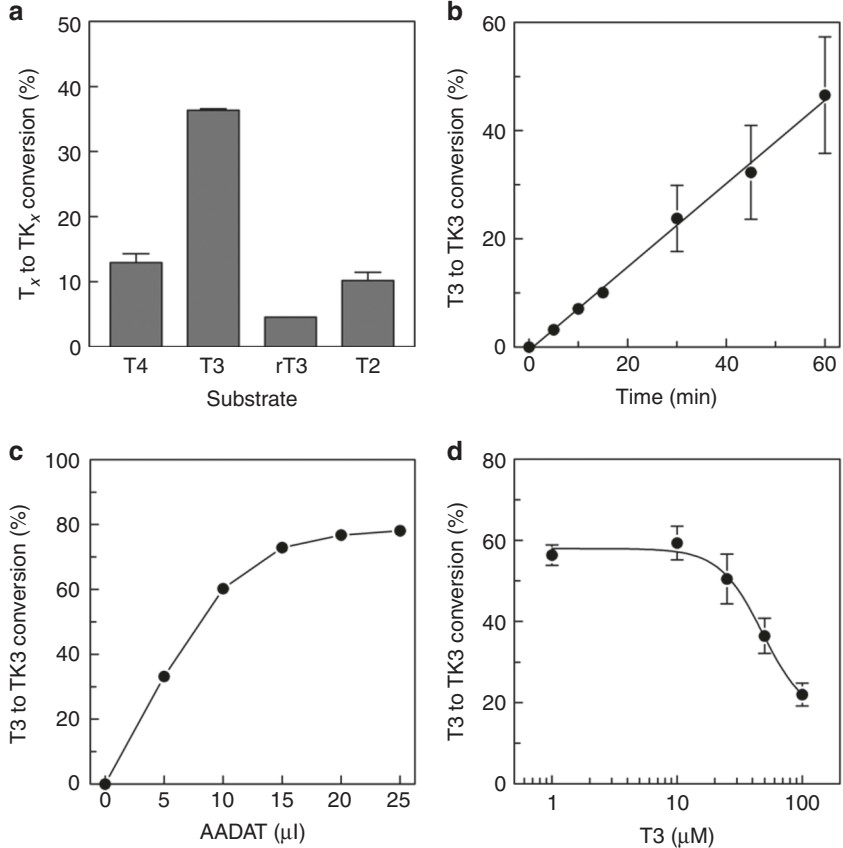

**Fig. 4** AADAT converts T3 and T4 to their respective pyruvic acid metabolites. **a** shows the conversion of T3 and T4 to their pyruvic acid metabolites TK3 and TK4 in cell lysates of hAADAT over-expressing COS-1 cells. Cell lysates were incubated with [I$^{125}$]-T3 or [I$^{125}$]-T4 ($2 \times 10^5$ c.p.m.) in the presence of 0.1 mM pyridoxal 5′-phosphate and 1 mM α-ketoglutaric acid for 30 min and the resulting radio-labeled metabolites were separated by UPLC. The conversion of T3 to TK3 depends on the incubation time (**b**) and amount of cell lysate added to the incubation reaction (**c**) and is saturated at substrate concentrations between 10 and 100 μM (**d**). The percentage conversion reflects the amount of TK3 as a percentage of the total radioactivity eluted from the UPLC column and is corrected for the TK3 production in lysates derived from empty-vector-transfected control cells (which was nearly absent). All data are presented as means ± SEM ($n = 3$)

independent TSH- and FT4-associated variants, respectively. Next, these GRSs were tested for association with the risk of hypothyroidism and hyperthyroidism in up to 21,287 individuals. Figure 5 shows substantial differences in the risk of thyroid dysfunction across the range of GRS scores. Individuals with a TSH-based GRS in the highest quartile compared to individuals with a GRS in the lowest quartile had an odds ratio of 2.53 ($p = 6.8 \times 10^{-32}$) for hypothyroidism and 0.19 ($p = 9.8 \times 10^{-31}$) for hyperthyroidism, respectively. Conversely, the FT4-based GRS did not show any significant associations with either hypo or hyperthyroidism (Supplementary Table 7), which is consistent with the limited overlap observed between TSH and FT4 loci.

A GRS using all TSH-associated variants showed a significant association with Graves' disease ($p = 2.9 \times 10^{-5}$) that remained significant after excluding the *PSORS1C1* variant ($p = 2.5 \times 10^{-4}$), indicating a polygenic contribution of TSH-associated variants detected in the general population to Graves' disease.

As normal thyroid function is essential for the physiological function of virtually all human tissues, we tested if the TSH and FT4 GRSs were associated with a broader range of phenotypes by using available GWAS results of these phenotypes. These results are shown in Supplementary Table 8 with effects provided per increase in standard deviation of either TSH or FT4. A higher TSH GRS was associated with both a lower risk of Graves' disease (odds ratio (OR) = 0.64, $p = 2.0 \times 10^{-5}$) and goiter (OR = 0.30, $p = 3.9 \times 10^{-27}$), and lower thyroid volume (Δvol = −23%, $p =$

$1.3 \times 10^{-37}$), whereas a higher FT4 GRS was associated with a higher risk of goiter (OR = 1.52, $p = 7.9 \times 10^{-3}$) and higher thyroid volume (Δvol = 9%, $p = 3.8 \times 10^{-3}$). In addition, a higher TSH GRS was associated with a lower risk of schizophrenia (OR = 0.94, $p = 0.01$), shorter height (sd[height]:beta = −0.05, $p = 2.0 \times 10^{-11}$), and reduced kidney function (ΔeGFR = −1%, $p = 1.4 \times 10^{-5}$), as well as higher LDL (sd[LDL]:beta = 0.04, $p = 4.9 \times 10^{-3}$) and total cholesterol levels (sd[chol]:beta = 0.05, $p = 1.1 \times 10^{-5}$). A higher FT4 GRS was additionally associated with taller height (sd[height]:beta = 0.04, $p = 2.9 \times 10^{-4}$), lower BMI (sd[BMI]:beta = −0.04, $p = 2.7 \times 10^{-3}$), and lower LDL (sd [LDL]:beta = −0.06, $p = 3.1 \times 10^{-4}$) and total cholesterol levels (sd[chol]:beta = −0.05, $p = 6.1 \times 10^{-3}$) (Supplementary Table 9). These associations match clinical and epidemiological observations.

## Discussion

With 8 million genetic variants tested in up to 72,167 individuals, we present results of the largest GWAS on thyroid function and dysfunction performed so far. We identified 109 significantly and independently associated genetic variants, doubling the number of loci known to regulate thyroid function, which explain a substantial part of the variation in these traits. Importantly, we detected associations between these variants and thyroid diseases as well as various clinical end points, and functionally

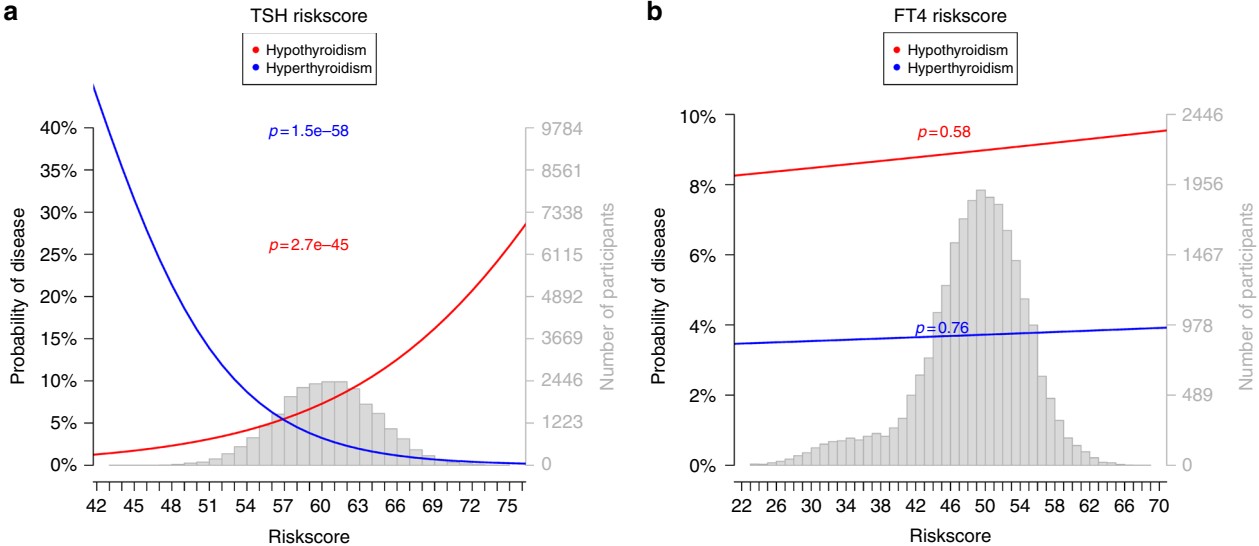

**Fig. 5** Associations of genetic risk scores with hypothyroidism and hyperthyroidism. The y axis shows the probability of hypothyroidism (red) or hyperthyroidism (blue) with the p-value of the association test of the trait on the risk score. The x axis shows the percentage of risk alleles carried based on a weighted genetic risk score (GRS) built using the 61 TSH-associated (**a**) and 31 FT4-associated GWAS SNPs (**b**). The gray histogram shows the distribution of the GRS in the study sample

characterized a new thyroid hormone transporter as well as a new thyroid hormone metabolizing enzyme.

Almost all previously identified TSH and FT4-associated SNPs were also genome-wide significantly associated with the respective trait in our analyses: the 20 TSH and 4 FT4 associations of the sex-combined GWAS of Porcu et al.[11], the two additional TSH SNPs as well as the FT4 association revealed in Taylor et al.[12], and 17 of the 21 genome-wide significantly TSH-associated SNPs identified by Gudmundsson et al.[13] The remaining loci of the latter study were discovered in our study via the mouse candidate analysis and were also associated with hypothyroidism (*TPO*), associated with FT4, hypo and hyperthyroidism (*FOXE1*), or had a $p < 1 \times 10^{-6}$ (*FOXE1, ELK3, SIVA1*). Only two FT4-associated loci, *LPCAT2/CAPNS2* and *NETO1/FBXO15*, identified in the sex-stratified analysis of Porcu et al. did not replicate in our sex-specific GWAS ($p \geq 0.01$). While all but 2 of the 18 cohorts (the Old Order Amish and the Baltimore longitudinal study on Aging) of Porcu et al. as well as four of the seven cohorts (TwinsUK GWAS, SardiNIA, Val Borbera and BHS) of Taylor et al. were also included in our GWAS, we more than doubled the sample size for both traits, thereby significantly increasing our power to detect new loci.

Consistent with HPT-axis physiology, most TSH-associated variants acted on gene expression levels in thyroid tissue, while the FT4-associated variants had more widespread effects on multiple known thyroid hormone target tissues. In addition, four of the newly identified variants were either associated with the risk of TPOAb positivity or Graves' disease, suggesting an underlying autoimmune-related pathophysiology. All of these findings confirm that GWAS in the general population provides a valuable method to identify genes implicated in thyroid physiology and/or thyroid disease. Moreover, these insights can be successfully translated into experimental evidence, as illustrated by our in vitro studies on *SLC17A4* and *AADAT*.

To investigate the combined effect of the thyroid hormone associated risk variants, we calculated a GRS. The GRS of TSH-associated variants was significantly associated with the risk of hypothyroidism and hyperthyroidism. For some FT4-associated hits, the lack of association with TSH levels can be explained on physiological grounds. For example, the identified SNP in *DIO1*

decreases the enzymatic activity of the protein, leading to less T4 to T3 conversion, resulting in higher T4 levels, but lower T3 levels, resulting in no net effect on feedback to the pituitary and therefore no effect on TSH levels. Similar hypotheses could be postulated for other loci involved in thyroid hormone metabolism, such as *DIO3OS* and *AADAT*. To assess whether these genetically estimated TSH levels are clinically relevant or merely reflect physiological inter-individual differences in TSH levels (i.e., HPT-axis setpoint), we tested the GRS against various clinical end points. These analyses showed significant associations with thyroid diseases, and also with altered lipid levels (total and LDL cholesterol) and height, which are both known to be affected by hypo and hyperthyroidism. For example, short stature is one of the key characteristics of patients suffering from congenital hypothyroidism. Interestingly, associations were also found with kidney function and schizophrenia, for which the causal relationships are less apparent. Thyroid hormone has been shown to influence kidney development and filtration function[37,38]. Likewise, rodent and human studies have shown that both hypo and hyperthyroidism lead to disrupted prenatal glial cell development, which is an important step in the development of schizophrenia[39], while various psychiatric diseases including schizophrenia are also thought to influence thyroid function via central effects on the HPT axis[40]. Future studies are needed to clarify the mechanisms underlying these associations, as it is not possible to solve these potential inverse causal relationships solely with GWAS results. Irrespective of the direction of the effects, our results suggest that the presence of kidney dysfunction and psychiatric symptoms in patients with thyroid disease deserve attention. Given the substantial increase in number of TSH and FT4-associated variants, which explain a substantial part of the variation in these traits, future studies should start exploring the use of these markers to predict the individual HPT-axis setpoint. This predicted setpoint could be used to guide treatment of thyroid diseases, which is important as despite normalized TSH and FT4 levels, a substantial part of treated patients still have persistent hypo or hyperthyroid complaints, leading to a lower quality of life[41]. This could be due to the fact that the TSH and FT4 levels are normalized to within the population-based reference ranges, but still deviate from the patient's individual

setpoint. For this purpose, a GWAS on the TSH/FT4 ratio could prove to be a more sensitive method to identify more variants, which specifically affect the HPT-axis setpoint.

When interpreting the results of our GWAS studies on increased and decreased TSH levels, it is important to realize that these studies were performed in population-based cohorts, and not in dedicated thyroid disease patient cohorts. Individuals on thyroid medication or a history of thyroid surgery were excluded, resulting in a relative overrepresentation of individuals with subclinical forms of thyroid dysfunction. The identified variants are therefore expected to be a mix of variants, which have been previously associated with hypo or hyperthyroidism (e.g., *TPO*, *FOXE1*, and *ATXN2*) and variants that lead to a TSH level, which is slightly above or below the population-based reference ranges. These latter effects can either reflect true mild thyroid dysfunction with increased risk of clinical consequences or merely reflect a deviation from the individual HPT-axis setpoint with no clinical consequences. While our GRS analyses suggest that carrying multiple risk alleles leads to an increased risk of overt thyroid dysfunction and related clinical consequences, the exact contribution of each individual variant needs to be clarified in future studies.

Thyroid hormone is importantly metabolized through enzymatic deiodination by DIO1-3, but also undergoes alternative metabolic reactions, including conjugation with sulfate or glucuronic acid and modification of the alanine side-chain. The latter includes the conversion of T3 and T4 to their respective pyruvic acid metabolites TK3 and TK4, which requires the oxidative deamination of their alanine side-chain[36]. TK3 and TK4 have been detected in urine and bile of rat injected with radio-labeled T3 and T4[42]. Although these and other studies suggested an important role for the liver and kidney in the formation of these pyruvic acid metabolites, the involved enzyme(s) had not been identified. Our functional analyses demonstrated that AADAT effectively catalyzes the transamination of T4 and in particular T3 to TK4 and TK3, respectively. Moreover, AADAT is highly expressed in the liver, gastrointestinal tract, and kidney in human[43]. Taken together, AADAT activity may thus be critical for the rate of thyroid hormone metabolism, which likely underlies the association of *AADAT* with circulating FT4. Although the specific impact of the associated variant on *AADAT* expression has not been assessed yet, our eQTL co-localization studies indicate that the index SNP decreases *AADAT* transcript levels in the thyroid, and this in turn leads to increased circulating FT4 levels.

The functional analyses further demonstrate that human SLC17A4 is able to transport both T4 and T3. The protein belongs to the solute carrier 17 family, whose members transport various organic anions, such as *p*-aminohippuric acid. Genetic variation in the *SLC17A4* locus has been associated with the progression of elevated serum urate levels to gout[27,44]. According to the GTEx data resource and previously reported studies[27], *SLC17A4* is predominantly expressed in human small intestinal and colonic epithelial cells, pancreas, liver, and kidney cortex, which could imply a role for this transporter in the metabolic clearance and entero-hepatic cycle of thyroid hormone. Future studies should investigate the pharmacokinetic properties of SLC17A4, its relative contributions to thyroid hormone transport in various individual tissues, as well as the effects of the identified *SLC17A4* variants on thyroid hormone transport.

The findings from our functional studies do not only provide new insights into thyroid hormone physiology, but may also have important clinical implications. Hypothyroidism is treated with levothyroxine (LT4), which is inexpensive and administered orally. In recent decades, various factors have been identified which help to determine LT4 dose, such as weight, gastrointestinal diseases, and interfering drugs[45,46]. Despite this knowledge, ineffective LT4 supplementation is still a major clinical problem, as 30–50% of patients are either under- or over-treated and therefore remain at risk for the symptoms and complications associated with thyroid dysfunction[45,46]. Therefore, the identification of SLC17A4 as a thyroid hormone transporter and AADAT as a thyroid hormone metabolizing enzyme provides new insights into thyroid hormone physiology and opens up a potential avenue for novel therapeutic targets or optimization of existing ones to improve the care of patients suffering from thyroid dysfunction. All genetic findings in our study were limited to common or low-frequency SNPs, whereas rare SNPs or structural variants may also contribute to the yet unexplained variance of thyroid function. Large whole-exome or -genome sequencing studies are required to reveal these rare variant associations[47]. Furthermore, additional GWAS with increased sample size will help to reveal the yet undiscovered associations of common and low-frequency SNPs. Our ThyroidOmics Consortium (http://www.thyroidomics.com) provides a well-established infrastructure to address these knowledge gaps in future projects.

## Methods

**Included studies.** Discovery meta-analyses included data from 22 independent cohorts with 54,288 subjects for the TSH analyses, and from 19 cohorts with 49,269 subjects for FT4, 53,423 subjects (3440 cases) for hypothyroidism, and 51,823 subjects (1840 cases) for hyperthyroidism (Supplementary Data 1). Selected SNPs from the TSH or FT4 analyses were carried forward for replication with in silico GWAS data from 5 cohorts (9053 subjects) and de novo genotyping in additional 5 cohorts (13,330 subjects). All subjects gave informed consent and studies were approved by the cohort-specific ethics committees.

We used the results of the GWAS of TPOAb positivity that included 18,297 subjects[20] for a look-up of all the 53 TSH-associated loci or their HapMapII proxies ($r^2 > 0.8$ in a 1 Mb window) that were available in that dataset to assess their relation to autoimmune hypothyroidism. A complementary look-up was performed for the 52 SNPs that were available in a GWAS on Graves' disease diagnosed by clinical examinations, circulating thyroid hormone and TSH concentrations, serum levels of antibodies against thyroglobulin, thyroid microsomes, and TSH receptors, ultrasonography, $^{[99m]}TCO_4^-$ (technetium-99m pertechnetate) (or $^{[123]}I$ (radioactive iodine)) uptake and thyroid scintigraphy using the data of the BioBank Japan Project (BBJ) including 1747 patients and 6420 controls (Supplementary Data 1).

**Trait definition.** In each study, only subjects with TSH levels within the cohort-specific reference range were included for the TSH and FT4 analyses. TSH and FT4 were analyzed as continuous variables after inverse normal transformation. Increased TSH was defined by a TSH level above the upper limit of the cohort-specific TSH reference range, while decreased TSH was defined by a level below the lower limit of the reference range. For both increased and decreased TSH analyses, the comparison group consisted of subject with a TSH level within the cohort-specific reference range. Exclusion criteria for all analyses were non-European ancestry, use of thyroid medication (defined as ATC (Anatomical Therapeutic Chemical) code H03), or previous thyroid surgery.

**GWAS in individual studies.** In each study of the discovery GWAS, genotyping was performed on genome-wide arrays. Genome-wide data were imputed to the 1000 Genomes, phase 1 version 3 (March 2012) ALL populations reference panel, including the X chromosome. Quality control before imputation was applied in each study separately. Details on study-specific genotyping and imputation information are provided in the Supplementary Data 1.

In the individual study GWAS, the association of the SNPs was analyzed using linear regression for TSH and FT4, and logistic regression for decreased and increased TSH. The genotype–phenotype association was conducted using an additive genetic model on SNP dosages, thus taking genotype uncertainties of imputed SNPs into account. The analyses for TSH and FT4 were initially sex-stratified and meta-analyzed as a second step. The analyses were adjusted for age, age-squared (to account for non-linear effects), and relevant study-specific covariates such as principal components for population stratification, study center, and family structure (e.g., by inclusion of the kinship matrix as a random effect), if applicable. The family-based cohorts GARP, SardiNIA, ValBorbera, MICROS, TwinsUK, LLS, and FHS conducted additional analyses on the men and women-combined sample, with additional adjustment for sex, to properly account for their family relatedness.

**Statistical methods for meta-analysis.** Result files from individual studies included in this analysis underwent extensive quality control before meta-analysis: file format checks as well as plausibility and distributions of association results including effects, standard errors, allele frequencies, and imputation quality of the

SNPs were obtained by using the gwasqc() function of the GWAtoolbox package v2.2.4[48]. Additionally, the known associations of rs6885099 in *PDE8B* with TSH and rs2235544 in *DIO1* with FT4 were checked for consistent effect direction and size in each study. All cohort-specific genomic control values ($\lambda_{GC}$) ranged from 0.94 to 1.14 (median 1.00) for the continuous trait and from 0.68 to 1.04 (median 0.91) for the dichotomous trait GWAS.

All meta-analyses were carried out in duplicate by three independent analysts. We conducted a fixed-effect meta-analysis applying inverse-variance weighting as implemented in Metal[49]. SNPs with MAF ≤0.005 or an imputation quality score ≤0.4 were excluded prior to the meta-analyses resulting in a median of 9,653,808 SNPs per cohort (IQR: 9,302,604–10,705,092). During the meta-analysis, the study-specific results were corrected by their specific $\lambda_{GC}$ if >1. Results were checked for possible errors like use of incorrect unit, trait transformation, or association model by plotting the association *p*-values of the analyses against those from a *z*-score-based meta-analysis for verifying overall concordance. SNPs that were present in <75% of the total sample size contributing to the respective meta-analysis (separately for autosomal and X-chromosomal SNPs) or with a MAF ≤0.01 (hypo and hyperthyroidism MAF ≤0.05 because of the low number of cases in the analysis) were excluded from subsequent analyses. Finally, data for up to 8,048,941 genotyped or imputed autosomal and X-chromosomal SNPs were available after the discovery stage meta-analysis of TSH, FT4, and up to 5,965,951 SNPs after hypo and hyperthyroidism.

Quantile–quantile plots of the meta-analysis results are provided in Supplementary Figures 10 and 11. To assess whether there was an inflation of *p*-values in the meta-analysis results attributed to reasons other than polygenicity, we performed LD score regression[50]. The LD score-corrected $\lambda_{GC}$ values of the meta-analysis results ranged from 1.00 to 1.04, supporting the absence of unaccounted population stratification. Genome-wide significance was defined as a *p*-value of $<5 \times 10^{-8}$, corresponding to a Bonferroni correction of one million independent tests. Unless stated otherwise, all reported *p*-values are two-sided. The $I^2$ statistic was used to evaluate between-study heterogeneity[51].

Gene-by-sex interaction on the circulating TSH and FT4 levels were obtained for each SNP by comparing the discovery meta-analysis results from men (TSH: $n = 24{,}618$; FT4 $n = 22{,}315$) and women using a *t*-test. Test statistics were calculated using the formula $t = (\beta_{men} - \beta_{women})/\mathrm{sqrt}(\mathrm{SE}_{men}^2 + \mathrm{SE}_{women}^2)$, assuming independent effect sizes between men and women.

To evaluate the presence of independent SNPs within each locus, SNPs were clustered based on their correlation with the SNP showing the lowest *p*-value at that locus (index SNP) using the software PLINK[52] and the genotypes of the combined individuals of the 1000Genomes phase1v3 all ethnicities reference panel (linkage disequilibrium pruning using $r^2 \leq 0.01$ within windows of ±1 Mb). The loci were named according to the nearest gene of the index SNP. Genomic positions correspond to build 37 (GRCh37).

**Replication analysis**. The genome-wide significant index SNPs of newly identified loci from the sex-combined TSH ($n = 22$) and FT4 meta-analyses ($n = 19$) were taken forward to the replication stage (Supplementary Table 10). When SNPs were not available in the in silico replication datasets, a proxy SNP in LD with $r^2 > 0.8$ was selected.

Of the ten studies that contributed to replication, five studies used 1000Genomes imputed dosages, three studies performed de novo genotyping, and two studies were genotyped on both the Illumina ExomeChip and CardiometaboChip. No SNP or proxy for the X-chromosomal locus was available in any replication dataset. The results from the discovery meta-analysis and the results of replication studies were meta-analyzed to obtain the overall *p*-values of the selected SNPs. SNPs with *p*-values below genome-wide significance in this combined analysis and with concordant effect directions in both stages were considered as replicated[53].

**Integration of information from genetically manipulated mice**. We tested whether information about thyroid function or disease from genetically manipulated mice could facilitate the detection of additional human thyroid loci that did not reach genome-wide significance in the GWAS (nested candidate gene approach). To this end, all genes that when manipulated cause abnormal thyroid physiology (MP:0002876; 26 genes) or abnormal thyroid gland morphology (MP:0000681; 51 genes) were selected from the comprehensive Mouse Genome Informatics resource in October of 2016 (http://www.informatics.jax.org/mp/). Next, genes were translated to their human homologs by the calculation of the number of independent SNPs in these genes with MAF > 0.01 in the 1000 Genomes EUR populations (PLINK option—indep-pairwise 50 5 0.2) to obtain multiple testing corrected significance thresholds ($p < 2.5 \times 10^{-5}$ for physiology and $p < 1.9 \times 10^{-5}$ for morphology). The genes in the respective mouse lists were then queried for the presence of SNPs that showed association with TSH and/or FT4 below the significance threshold. To test whether the number of genes with significant associations was higher than expected by chance, results were compared to those from 2000 iterations of random gene lists of equal length. A *p*-value for enrichment was computed from a complementary cumulative binomial distribution as described in detail previously[54]. Lastly, novel loci that were not identified at genome-wide significance in the GWAS of TSH or FT4 were tested for replication in up to 9011 and 4532 independent samples for TSH and FT4, respectively.

Successful replication was defined as direction-consistent association and genome-wide significance in a meta-analysis of the discovery and replication samples.

**eQTL look-up**. To assess the possible effect of our lead signals on transcriptional activity, we queried expression QTL (eQTL) results from 22 publicly available studies (specific reference listed in Supplementary Data 3). These studies were carried out from 2007 to February 2017 on 127 different tissues and cell types, and used either micro-array or sequencing-based assessment of gene expression. For each study, we derived the list of top eQTLs by LD clumping, and searched top eQTLs in high LD ($r^2 > 0.8$ in 1000Genomes EUR samples) with the 94 thyroid function-associated index or independent SNPs (TSH, FT4, and *ATXN2* and *TPO* as additional GWAS loci from hypothyroidism) (Supplementary Tables 1, 2, 4).

To evaluate the evidence of co-localization between the index GWAS and eQTL SNPs, we used the SMR method[24], coupled with the test for heterogeneity of effects (HEIDI)[24]. The first tests whether the effect on expression seen at a SNP or at its proxies correlates with the signal observed in the GWAS (SMR test), while the second evaluates if the eQTL and GWAS associations can be attributable to the same causative variant (HEIDI test). Because direction of effects has to be taken into account, we focused this analysis only on GTEx data for which full summary results were available. For SMR, we considered the experiment-wise *p*-value of $2 \times 10^{-4}$ (corresponding to a Bonferroni correction for 242 gene-thyroid trait-tissue combinations assessed). Specifically, we tested all genes with an eQTL *p*-value $<1 \times 10^{-7}$ and for which the top eQTL showed genome-wide significant association with any thyroid hormone traits, regardless of LD between the top eQTL and the thyroid hormone index SNP. For the HEIDI test, we used the suggested *p*-value >0.05 cutoff to declare co-localization, and further required that at least five SNPs were available for the test[24].

**Materials for in vitro studies**. [$^{125}$I]T3 and [$^{125}$I]T4 were synthesized using the standard chloramine-T method[55]. Unlabeled iodothyronines, pyridoxal 5′-phosphate (PLP), 4-(2-hydroxyethyl)-1-piperazineethanesulfonic acid (HEPES), bovine serum albumin, D-glucose, and Na$_2$SeO$_3$ were obtained from Sigma-Aldrich (Zwijndrecht, The Netherlands); and α-ketoglutaric acid (KG) from Merck Millipore (Amsterdam, NL).

**Expression constructs and cloning**. The cDNA of MCT8 and CRYM was cloned into pcDNA3 and pSG5 expression vectors, respectively, using standard cloning techniques[56,57]. A pCMV6-Entry_SLC17A4 expression vector containing a C-terminal Myc and Flag tag was obtained from OriGene Technologies (Rockville, USA). A pbluescript AADAT cDNA construct was obtained from Thermo Scientific (Bleiswijk, NL) and subcloned into pcDNA3 with addition of a C-terminal Flag-tag using standard cloning techniques. Any variants were substituted in agreement with the NM_001286683.1 reference sequence using Quikchange site-directed mutagenesis according to manufacturer's protocol (Stratagene, Amsterdam, The Netherlands). All primers are available upon request. Correctness of all expression constructs was confirmed by sequencing of the inserts.

**Cell culture and transfection**. COS-1 African green monkey kidney cells were obtained from ECACC (Sigma-Aldrich, Zwijndrecht, NL) and cultured in DMEM/F12 (Life Technologies, Bleiswijk, NL) containing 9% heat-inactivated fetal bovine serum (Sigma-Aldrich) and 0.2 mg mL$^{-1}$ penicillin/streptomycin (Life Technologies). Cell culture flasks and dishes were obtained from Corning (Schiphol, NL).

For T3 and T4 uptake studies, COS-1 cells were seeded in 24-well dishes ($1 \times 10^5$ cells per well) and transiently transfected at 70% confluence with 250 ng empty vector (EV), SLC17A4, or MCT8, with or without the addition of 100 ng CRYM. CRYM is a cytoplasmic high-affinity thyroid hormone-binding protein, which prevents efflux of internalized thyroid hormones. For thyroid hormone metabolism assays, COS-1 cells were seeded on 10 cm dishes ($5 \times 10^5$ cells per well) and transiently transfected with 2000 ng EV or AADAT at 70% confluence. Xtreme-Gene 9 was used as a transfection reagent according to manufacturer's protocol (Roche Diagnostics, Almere, NL).

**Thyroid hormone uptake studies**. Thyroid hormone uptake studies were performed according to well-established protocols[58,59]. Cells were washed with incubation medium (Dulbecco's PBS (D-PBS) and 0.1% D-glucose) and incubated for 10 min with 1 nM ($5 \times 10^4$ c.p.m.) [$^{125}$I]T$_3$ or [$^{125}$I]T$_4$ in 375 μl incubation medium at 37 °C. Finally, cells were washed once with incubation medium and lysed with 0.1 M NaOH. Radioactivity in the lysates was measured with a γ-counter. For saturation experiments, the indicated concentrations of unlabeled T3 or T4 were added to the incubation medium.

**Thyroid hormone metabolism studies**. Two days after transfection, cells were harvested in 20 mM HEPES buffer (pH 7.5) and lysed by vortexing the sample for 30 s. Samples were clarified by centrifugation at 20,000×*g* for 30 min at 4 °C. Thyroid hormone aminotransferase activity was measured in duplicate by

incubating 0.1-100 μM of T3 or T4 in the presence of $2 \times 10^5$ c.p.m. $^{125}$I-labled hormone for 5–60 min at 37 °C with 5–25 μl clarified lysate, 1 mM KG and 0.1 mM PLP in a total volume of 100 μl HEPES buffer. Reactions were quenched by addition of 125 μl ice-cold 0.1% acetic acid in acetonitrile, followed by 1 h of incubation on ice to precipitate proteins. After centrifugation (20,000×$g$, 15 min, 4 °C), 125 μl of the supernatant was mixed with 125 μl ammonium acetate buffer (20 mM, pH 4.0), and 100 μl was analyzed by UPLC (Waters, Etten-Leur, NL) using a BEH C18 reversed phase column (130 Å, 2.1 × 100 mm, 1.7 μm). Ammonium acetate buffer (20 mM, pH 4.0, solvent A) and 0.1% acetic acid in acetonitrile (solvent B) were used as mobile phase. Flow rate was 0.35 mL min$^{-1}$, and column temperature 30 °C. The gradient used was: 0–1 min (30% B), 1–7 min (30–42% B), 7–23 min (42–58% B), 25–27 min (58–100% B), and 27–32 min (100–30% B). The radioactivity in the eluate was monitored using a Radiomatic A-500 flow scintillation detector (Packard Instruments, Meriden, CT).

**Genetic risk score analysis.** Two separate GWAS effect size-weighted GRS were generated to evaluate the combined effect of the TSH- and FT4-increasing alleles, respectively, on the risk of hypo and hyperthyroidism using individual level data from four of our largest GWAS studies: ARIC, CHS, Rotterdam Study, and SHIP (hypothyroidism cases: $n = 1613$; hyperthyroidism cases: $n = 662$; controls: $n = 19,674$). The GRS were based on the 61 and 31 replicated GWAS SNPs for TSH and FT4, respectively (Supplementary Tables 1 and 2), normalized to a range of 0 to 100, associated in each cohort separately using a logistic regression adjusted for sex and age, and combined afterwards by a fixed-effect inverse-variance meta-analysis using R[60]. The probability of disease was calculated using the formula $1/(1 + \exp(-(\beta_0 + \beta_1*x)))$, where $\beta_0$ and $\beta_1$ correspond to the intercept and GRS-related effect in the regression model, respectively.

To test the combined effect of the replicated TSH and FT4 SNPs on various traits (Supplementary Tables 8 and 9), a GRS-based association on meta-analyses results was performed as described in reference[61] using the function grs.summary() of the R-package gtx. If a specific SNP was not available in the look-up GWAS dataset, a proxy SNP in LD with $r^2 > 0.8$ was included where possible. The association effects correspond to a change in the look-up GWAS trait (or natural logarithm of the odds ratio in the case of a binary trait) per standard deviation unit of TSH and FT4, respectively. In case the trait was logarithm-transformed in the GWAS, $(e^{\text{beta}} - 1) \times 100$ corresponds to a 1% change of this trait.

**Pathway analyses.** We performed Data-Driven Expression Prioritized Integration for Complex Traits (DEPICT)[62] analyses to prioritize the most likely causal genes at the associated loci and identify enriched pathways and tissues. We used DEPICT version 1 rel194 and included variants with GC-corrected $p$-value $<1 \times 10^{-5}$ from discovery GWAS as input. The following parameters were applied in the DEPICT analyses: 50 repetitions used to compute the false discovery rate, 500 permutations used to adjust for biases such as gene length, and 500 null GWAS used to run repetitions and permutations.

Genes for network analysis were selected using the associated genes from DEPICT using the lead SNPs from the analyses of the discovery GWAS. The Ingenuity Pathway Analysis Software Tool (IPA; Ingenuity® Systems, CA, USA) Network was used in order to perform pathway analysis (core analysis). Molecules and/or relationships considered were the ones available in the IPA Knowledge Base for mammals. Confidence filters considered only relationships, direct and indirect, where the confidence is experimentally observed or high (predicted). Networks were generated with a maximum size of 35 genes and allowing up to 25 networks per analysis. We did not restrict for tissue and cell lines or mutations. The networks are constructed using the IPA algorithm with the Ingenuity Knowledge Base as a reference set generating a score as well as a $p$-value. IPA computes a score for each network according to the fit of that network to the user-defined set of focus genes. The score indicates the likelihood of the focus genes in a network being found together due to random chance. The significance of $p$-value is calculated using the right-tailed fisher exact test.

**Look-up of pleiotropy.** The look-up of additional traits associated with the replicated GWAS findings were performed using the PhenoScanner[63] database with the default search options, whereas the independent SNPs of the replicated TSH, FT4, hypo, and hyperthyroidism-associated loci or their proxies ($r^2 > 0.8$) obtained by SNiPa[64] were used as input. Look-up results with genome-wide significant $p$-value ($5 \times 10^{-8}$) were reported.

## Data availability

Summary genetic association results are available for full download or visualization on the CHARGE dbGaP website under accession phs000930 (https://www.ncbi.nlm.nih.gov/gap) and in the locuszoom web page: http://locuszoom.sph.umich.edu/genform.php.

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

## Acknowledgements

This manuscript is dedicated to the memory of Professor Dr. Theo J. Visser, a brilliant scientist and mentor, who recently passed away. The project was conducted within the ThyroidOmics Consortium (http://www.thyroidomics.com). Extended acknowledgments and study-specific acknowledgments are provided in Supplementary Note 4.

## Author contributions

Design, management, or subject recruitment of the individual studies: A.A., A.A.H., A.Astrup, A.C., A.D.G., A.G.U., A.K., A.L., A.P., A.T., B.L., B.M.P., B.T., C.C., C.D.M., C.F.S., C.G., C.M., C.P., D.J.S., D.M., D.S., D.Tiller, D.Toniolo, D.v.H., E.F., E.K., E.M.L., E.R.R., E.S., F.C., F.K., F.R., G.C., G.R., H.V., I.F., I.J.D., I.M., J.G.E., J.K., J.L., J.M.S., J.P.W., J.W.J., K.E., K.H.G., K.M., L.A.K., L.C.S., L.F., M.A.I., M.B., M.D.B., M.G., M.K., M.Kubo, M.M., M.Medici, M.d.H., O.P., P.E.S., P.P.P., P.R., R.M., R.P.P., R.T.N., S.B., S.E.H., S.G., S.G.W., S.N., S.S., T.D.M., T.D.S., T.E.G., T.F., T.H., T.I.A.S., T.J.V., and Y.T. Drafting of manuscript: A.C., A.K., A.T., L.C., M.Medici, R.P.P., S.G., S.N., S.S., T.J.V., U.T.S., and Y.L. Project design, analysis, or interpretation of the results: A.K., A.T., C.B., E.Kasbohm, H.V., L.C., M.Medici, M.T., R.P.P., S.S., T.J.V., U.T.S., and Y.L. Genotyping or phenotyping of the individual studies: A.B.E., A.G.U., A.K., B.H.W., B.L., B.M.P., C.F.S., C.G., D.C.M.L., E.B., E.M.L., E.S., E.V.R.A., F.R., G.H., G.R., H.M.z.S., H.Z., I.M., J.B.R., J.D., J.H., J.I.R., J.P.B., J.P.W., J.W.J., L.C.S., M.B., M.Kubo, N.G., N.S., O.P., R.R., S.E.H., S.G.W., T.D.S., T.F., T.H., T.I.A.S., T.T., W.d.H., and Y.F.M.R. Functional studies: C.D.M., S.G., and T.J.V. Statistical methods, analysis, bioinformatics, or interpretation of the results in the individual studies: A.A., A.K., A.L., A.T., B.T., C.B., C.C., C.D.M., C.F., C.F.S., C.P., D.E.A., D.M., D.Medenwald, D.T., D.Tiller, E.Kasbohm, E.M.L., E.P., E.R.R., W.E.V., E.V.R.A., G.R., H.M.z.S., I.F., I.M., I.M.N., I.P., J.D., J.J., J.L., J.P.W., K.H., K.H.G., L.C., M.A., M.A.I., M.D.B., M.Medici, M.Moed, M.P., M.T., N.G., P.J.C., Q.Y., R.A.J., R.M., R.P.P., R.R., S.B., S.E.H., S.G., S.G.W., S.J.B., S.L., S.N., S.S., T.D.M., T.E.G., T.I., T.J.V., T.S.A., T.T., U.T.S., W.d.H., Y.K., Y.L., Y.O., Y.F.M.R., and Y.T. Critical review of manuscript: all authors.

## Additional information

**Competing interests:** B.M.P. serves on the DSMB of a clinical trial funded by Zoll LifeCor and on the Steering Committee of the Yale Open Data Access Project funded by Johnson & Johnson. The remaining authors declare no competing interests.

Alexander Teumer[1,2], Layal Chaker[3,4,5], Stefan Groeneweg[3,5], Yong Li[6], Celia Di Munno[3,7], Caterina Barbieri[8], Ulla T. Schultheiss[6], Michela Traglia[8], Tarunveer S. Ahluwalia[9,10], Masato Akiyama[11], Emil Vincent R. Appel[10], Dan E. Arking[12], Alice Arnold[13], Arne Astrup[14], Marian Beekman[15], John P. Beilby[16,17], Sofie Bekaert[18], Eric Boerwinkle[19], Suzanne J. Brown[20], Marc De Buyzere[21], Purdey J. Campbell[20], Graziano Ceresini[22], Charlotte Cerqueira[23], Francesco Cucca[24,25], Ian J. Deary[26,27], Joris Deelen[15,28], Kai-Uwe Eckardt[29,30], Arif B. Ekici[31], Johan G. Eriksson[32,33,34], Luigi Ferrrucci[35], Tom Fiers[36], Edoardo Fiorillo[24], Ian Ford[37], Caroline S. Fox[38,39], Christian Fuchsberger[40], Tessel E. Galesloot[41], Christian Gieger[42,43], Martin Gögele[40], Alessandro De Grandi[40], Niels Grarup[10], Karin Halina Greiser[44], Kadri Haljas[45], Torben Hansen[10], Sarah E. Harris[26,46], Diana van Heemst[47], Martin den Heijer[48], Andrew A. Hicks[40], Wouter den Hollander[49], Georg Homuth[50], Jennie Hui[16,51], M. Arfan Ikram[4], Till Ittermann[1,2], Richard A. Jensen[52], Jiaojiao Jing[6,53], J. Wouter Jukema[54,55], Eero Kajantie[34,56,57], Yoichiro Kamatani[11,58], Elisa Kasbohm[1,59], Jean-Marc Kaufman[60], Lambertus A. Kiemeney[41], Margreet Kloppenburg[61,62], Florian Kronenberg[63], Michiaki Kubo[64], Jari Lahti[45], Bruno Lapauw[60], Shuo Li[65], David C.M. Liewald[26,27], Lifelines Cohort Study, Ee Mun Lim[16,20], Allan Linneberg[23,66], Michela Marina[22], Deborah Mascalzoni[40], Koichi Matsuda[67], Daniel Medenwald[68], Christa Meisinger[42,69], Ingrid Meulenbelt[49], Tim De Meyer[70], Henriette E. Meyer zu Schwabedissen[71], Rafael Mikolajczyk[68], Matthijs Moed[15], Romana T. Netea-Maier[72], Ilja M. Nolte[73], Yukinori Okada[11,74,75], Mauro Pala[24], Cristian Pattaro[40], Oluf Pedersen[10], Astrid Petersmann[76], Eleonora Porcu[24,77], Iris Postmus[78], Peter P. Pramstaller[40], Bruce M. Psaty[79,80], Yolande F.M. Ramos[49], Rajesh Rawal[42], Paul Redmond[27], J. Brent Richards[81,82], Ernst R. Rietzschel[83,84], Fernando Rivadeneira[4,5], Greet Roef[60], Jerome I. Rotter[85], Cinzia F. Sala[8], David Schlessinger[86], Elizabeth Selvin[87], P. Eline Slagboom[15], Nicole Soranzo[88], Thorkild I.A. Sørensen[10,89], Timothy D. Spector[82], John M. Starr[26,90], David J. Stott[91], Youri Taes[92], Daniel Taliun[40], Toshiko Tanaka[35], Betina Thuesen[23], Daniel Tiller[69], Daniela Toniolo[8], Andre G. Uitterlinden[4,5], W. Edward Visser[3,5], John P. Walsh[20,51], Scott G. Wilson[20,51,82], Bruce H.R. Wolffenbuttel[93], Qiong Yang[65], Hou-Feng Zheng[94,95], Anne Cappola[96], Robin P. Peeters[3,5], Silvia Naitza[24], Henry Völzke[1,2], Serena Sanna[24,97], Anna Köttgen[6,98], Theo J. Visser[3,5] & Marco Medici[3,4,5]

[1]Institute for Community Medicine, University Medicine Greifswald, Greifswald, Germany. [2]DZHK (German Center for Cardiovascular Research), Partner Site Greifswald, Greifswald, Germany. [3]Erasmus MC Academic Center for Thyroid Diseases, Rotterdam, The Netherlands. [4]Department of Epidemiology, Erasmus Medical Center, Rotterdam, The Netherlands. [5]Department of Internal Medicine, Erasmus Medical Center, Rotterdam, The Netherlands. [6]Department of Biometry, Epidemiology and Medical Bioinformatics, Institute of Genetic Epidemiology, Faculty of Medicine and Medical Center—University of Freiburg, Freiburg, Germany. [7]Department of Science and Technologies, University of Sannio, Benevento, Italy. [8]Division of Genetics and Cell Biology, San Raffaele Scientific Institute, Milan, Italy. [9]Steno Diabetes Center Copenhagen, Gentofte, Denmark. [10]Novo Nordisk Foundation Center for Basic Metabolic Research, Faculty of Health and Medical Sciences, University of Copenhagen, Copenhagen, Denmark. [11]Laboratory for Statistical Analysis, RIKEN Center for Integrative Medical Sciences, Yokohama, Japan. [12]McKusick-Nathans Institute of Genetic Medicine, Johns Hopkins University School of Medicine, Baltimore, MD, USA. [13]Department of Biostatistics, University of Washington, Seattle, WA, USA. [14]Department of Nutrition, Exercise, and Sports, Faculty of Science, University of Copenhagen, Copenhagen, Denmark. [15]Molecular Epidemiology, Department of Biomedical Data Sciences, Leiden University Medical Center, Leiden, The Netherlands. [16]Pathwest Laboratory Medicine WA, Nedlands, WA 6009, Australia. [17]School of Biomedical Sciences, University of Western Australia, Crawley, WA 6009, Australia. [18]Bimetra, Clinical Research Center Ghent, Ghent University Hospital, Ghent, Belgium. [19]Human Genetics Center, University of Texas Health Science Center, 1200 Pressler Street, Houston, TX 77030, USA. [20]Department of Endocrinology and Diabetes, Sir Charles Gairdner Hospital, Nedlands, WA 6009, Australia. [21]Department of Cardiology, Ghent University Hospital, Ghent, Belgium. [22]Department of Medicine and Surgery, University of Parma, University Hospital of Parma, Parma, Italy. [23]Center for Clinical Research and Prevention, Bispebjerg and Frederiksberg Hospital, The Capital Region, Copenhagen, Denmark. [24]Istituto di Ricerca Genetica e Biomedica, Consiglio Nazionale delle Ricerche Monserrato, Monserrato, Italy. [25]Dipartimento di Scienze Biomediche, Università degli Studi di Sassari, Sassari, Italy. [26]Centre for Cognitive Ageing and Cognitive Epidemiology, University of Edinburgh, Edinburgh, UK. [27]Department of Psychology, University of Edinburgh, Edinburgh, UK. [28]Max Planck Institute for Biology of Ageing, Cologne, Germany. [29]Department of Nephrology and Hypertension, University of Erlangen-Nürnberg, Erlangen, Germany. [30]Department of Nephrology and Medical Intensive Care, Charité – Universitätsmedizin Berlin, Berlin, Germany. [31]Institute of Human Genetics, Friedrich-Alexander-Universität Erlangen-Nürnberg (FAU), Erlangen, Germany. [32]Department of General Practice and Primary Health Care, University of Helsinki and Helsinki University Hospital, Helsinki, Finland. [33]Folkhälsan Research Center, Helsinki, Finland. [34]National Institute for Health and Welfare, Helsinki, Finland. [35]Translational Gerontology Branch, National Institute on Aging, Baltimore, MD, USA. [36]Department of Clinical Chemistry, Ghent University Hospital, Ghent, Belgium. [37]Robertson Centre for Biostatistics, University of Glasgow, Glasgow, UK. [38]National

Heart, Lung, and Blood Institute's Framingham Heart Study and the Center for Population Studies, Framingham, MA, USA. [39]Division of Endocrinology, Brigham and Women's Hospital and Harvard Medical School, Boston, MA, USA. [40]Institute for Biomedicine, Eurac Research, Affiliated Institute of the University of Lubeck, Bolzano, Italy. [41]Radboud University Medical Center, Radboud Institute for Health Sciences, Nijmegen, The Netherlands. [42]Helmholtz Zentrum München, Institute of Epidemiology, German Research Center for Environmental Health, Neuherberg, Germany. [43]Research Unit of Molecular Epidemiology, Helmholtz Zentrum München, German Research Centre for Environmental Health, Neuherberg, Germany. [44]German Research Centre Division of Cancer Epidemiology, Heidelberg, Germany. [45]Department of Psychology and Logopedics, Faculty of Medicine, University of Helsinki, Helsinki, Finland. [46]Medical Genetics Section, University of Edinburgh Centre for Genomic and Experimental Medicine and MRC Institute of Genetics and Molecular Medicine, Edinburgh, UK. [47]Leiden University Medical Center, Geriatrics and Gerontology, Leiden, The Netherlands. [48]Vrije Universiteit Medisch Centrum, Amsterdam, The Netherlands. [49]Department of Biomedical Data Sciences, Section of Molecular Epidemiology, Leiden University Medical Center, Leiden, The Netherlands. [50]Interfaculty Institute for Genetics and Functional Genomics, University Medicine Greifswald, Greifswald, Germany. [51]Medical School, University of Western Australia, Crawley, WA 6009, Australia. [52]Department of Medicine, Cardiovascular Health Research Unit, University of Washington, Seattle, WA, USA. [53]Faculty of Biology, University of Freiburg, Freiburg, Germany. [54]Department of Cardiology, Leiden University Medical Center, Leiden, The Netherlands. [55]Einthoven Laboratory for Experimental Vascular Medicine, LUMC, Leiden, The Netherlands. [56]Children's Hospital, Helsinki University Hospital and University of Helsinki, Helsinki, Finland. [57]PEDEGO Research Unit, MRC Oulu, Oulu University Hospital and University of Oulu, Oulu, Finland. [58]Center for Genomic Medicine, Kyoto University Graduate School of Medicine, Kyoto, Japan. [59]Institute of Epidemiology, Friedrich-Loeffler-Institut (FLI), Federal Research Institute for Animal Health, Greifswald, Germany. [60]Department of Endocrinology, Ghent University Hospital, Ghent, Belgium. [61]Department of Rheumatology, Leiden University Medical Center, Leiden, The Netherlands. [62]Department of Clinical Epidemiology, Leiden University Medical Center, Leiden, The Netherlands. [63]Department of Medical Genetics, Division of Genetic Epidemiology, Molecular and Clinical Pharmacology, Medical University of Innsbruck, Innsbruck, Austria. [64]Laboratory for Genotyping Development, RIKEN Center for Integrative Medical Sciences, Yokohama, Japan. [65]Department of Biostatistics, Boston University, Boston, MA, USA. [66]Department of Clinical Experimental Research, Rigshospitalet, Copenhagen, Denmark. [67]Department of Computational Biology, Graduate School of Frontier Sciences, The University of Tokyo, Tokyo, Japan. [68]Institute for Medical Epidemiology, Biostatistics and Informatics, Martin-Luther-University Halle-Wittenberg, Halle, Germany. [69]Chair of Epidemiology, Ludwig-Maximilians-Universität München, at UNIKA-T Augsburg, Augsburg, Germany. [70]Department of Data Analysis and Mathematical Modelling, Ghent University, Ghent, Belgium. [71]Department of Pharmaceutical Sciences, University of Basel, Basel, Switzerland. [72]Department of Internal Medicine, Division of Endocrinology, Radboud University Medical Center, Nijmegen, The Netherlands. [73]Department of Epidemiology, University of Groningen, University Medical Center Groningen, Groningen, The Netherlands. [74]Department of Statistical Genetics, Osaka University Graduate School of Medicine, Osaka, Japan. [75]Laboratory of Statistical Immunology, Immunology Frontier Research Center (WPI-IFReC), Osaka University, Suita, Japan. [76]Institute of Clinical Chemistry and Laboratory Medicine, University Medicine Greifswald, Greifswald, Germany. [77]Center for Integrative Genomics, University of Lausanne, and Swiss Institute of Bioinformatics, 1015 Lausanne, Switzerland. [78]Department of Internal Medicine, Section of Gerontology and Geriatrics, Leiden University Medical Center, Leiden, Netherlands. [79]Cardiovascular Health Research Unit, Departments of Medicine, Epidemiology and Health Services, University of Washington, Seattle, WA, USA. [80]Kaiser Permanente Washington Health Research Institute, Seattle, WA, USA. [81]Lady Davis Institute, Jewish General Hospital, Montreal, QC H3T 1E2, Canada. [82]The Department of Twin Research and Genetic Epidemiology, King's College London, St. Thomas' Campus, Lambeth Palace Road, London SE1 7EH, UK. [83]Biobanking and Cardiovascular Epidemiology, Ghent University Hospital, Ghent, Belgium. [84]Department of Internal Medicine (Cardiology), Ghent University, Ghent, Belgium. [85]The Institute for Translational Genomics and Population Sciences, Departments of Pediatrics and Medicine, LABioMed at Harbor-UCLA Medical Center, Torrance, CA, USA. [86]Laboratory of Genetics and Genomics, National Institute on Aging, National Institutes of Health, Baltimore, MD, USA. [87]Department of Epidemiology, Welch Center for Prevention, Epidemiology and Clinical Research, Johns Hopkins Bloomberg School of Public Health, Baltimore, MD, USA. [88]The Wellcome Trust Sanger Institute, Wellcome Trust Genome Campus, Hinxton, Cambridge CB10 1HH, UK. [89]Department of Public Health (Section on Epidemiology), Faculty of Health and Medical Sciences, University of Copenhagen, Copenhagen, Denmark. [90]Alzheimer Scotland Dementia Research Centre, University of Edinburgh, Edinburgh, UK. [91]Institute of Cardiovascular and Medical Sciences, Faculty of Medicine, University of Glasgow, Glasgow, UK. [92]AZ Sint Jan Brugge, Brugge, Belgium. [93]Department of Endocrinology, University of Groningen, University Medical Center Groningen, Groningen, The Netherlands. [94]Diseases & Population (DaP) Geninfo Lab, School of Life Sciences, Westlake University and Westlake Institute for Advanced Study, Hangzhou, Zhejiang, China. [95]Institute of Aging Research and the Affiliated Hospital, School of Medicine, Hangzhou Normal University, Hangzhou, Zhejiang, China. [96]Division of Endocrinology, Diabetes, and Metabolism, University of Pennsylvania School of Medicine, Philadelphia, PA, USA. [97]Department of Genetics, University of Groningen, University Medical Center Groningen, Groningen, The Netherlands. [98]Department of Epidemiology, Johns Hopkins Bloomberg School of Public Health, 625 N Wolfe Street, Baltimore, MD 21205, USA. These authors contributed equally: Alexander Teumer, Layal Chaker, Stefan Groeneweg, Yong Li, Celia Di Munno, Caterina Barbieri. These authors jointly supervised this work: Henry Völzke, Serena Sanna, Anna Köttgen, Theo J. Visser, Marco Medici. Deceased: Theo J. Visser. A full list of consortium members appears below.

## Lifelines Cohort Study

Behrooz Z. Alizadeh[99], H. Marike Boezen[99], Lude Franke[100], Pim van der Harst[101], Gerjan Navis[102], Marianne Rots[103], Harold Snieder[99], Morris A. Swertz ![ORCID][100] & Cisca Wijmenga[100]

[99]Department of Epidemiology, University of Groningen, University Medical Center Groningen, Groningen, The Netherlands. [100]Department of Genetics, University of Groningen, University Medical Center Groningen, Groningen, The Netherlands. [101]Department of Cardiology, University of Groningen, University Medical Center Groningen, Groningen, The Netherlands. [102]Department of Internal Medicine, Division of Nephrology, University of Groningen, University Medical Center Groningen, Groningen, The Netherlands. [103]Department of Pathology and Medical Biology, University of Groningen, University Medical Center Groningen, Groningen, The Netherlands

