## [Peer Review File · Nature Communications]

Reviewer #1 (Remarks to the Author):

This is an interesting study arising from a large collaboration of established experts in the field. Given thyroid disease is common and the consequences of thyroid disease can influence key health outcomes including ischaemic heart disease and osteoporosis, this is of interest to individuals outside of thyroidology. This paper builds on earlier work by Porcu et al. and Taylor et al. and provides a substantial increase in SNPs identified and the improvements to our knowledge of the genetic architecture of thyroid function. The manuscript is well written and analyses are appropriate and the paper is likely to be widely cited. I have no major comments.

Introduction

Well written only a very minor point in that variation in thyroid even with the population reference range is associated with adverse outcomes, not just subclinical overt disease.

Methods and Results

Methods are clearly described

Results are well presented and figures and tables are clear and add to the text

Discussion

A few points to consider to enhance the manuscript for readability

Given the extra variation in TSH explained, would it be potentially possible to identify to an extent what an individual's target TSH on levothyroxine is most likely to be based on their , could this be discussed in more detail ?

Whilst a substantial increase has been explained in the genetic architecture of thyroid function, there is still some "dark matter" would the authors speculate on where to find this, do we need even larger GWAS or should we be focussing on rare variants (MAF <1%)

Given the size of the study where the authors surprised that variants in THRB were not associated with changes in thyroid status were they surprised not to see any other genes

Reviewer #2 (Remarks to the Author):

Teumer et al. conducted a meta-analysis of GWAS on TSH and FT4 levels. In the discovery stage that included 54,288 individuals from 22 cohort studies, they reported 22 SNPs significantly associated to TSH levels and 19 SNPs associated to FT4 levels that were not reported in previous studies. Of these, they replicated 19 SNPs associated to TSH and 16 SNPs associated to FT4 levels in 10 replication cohorts (5 with imputed data n=9053 and 5 with genotyped data n=13330).

Then they conducted several secondary analyses:

- (1) They selected candidate gene from a mouse model (26 genes), associated to mice abnormal physiology or morphology (51 genes). They found that variants from 4 of these genes were associated to TSH and FT4 levels in the discovery phase with p-values at the limit of the significance and 2 of these variants in genes CGA and TPO were replicated using the replication cohorts.
- (2) Using the same data, they defined hypothyroidism and hyperthyroidism using TSH levels, and then they conducted a GWAS study for these traits using logistic regression. They identified 7 susceptibility loci for hypothyroidism and 8 susceptibility loci for hyperthyroidism. Some of these loci were reported previously, but not all.
- (3) They defined genetic risk scores (GRS) for TSH levels and for FT4 levels and analyze the association with other phenotypes. They reported association with several traits including hypothyroidism, hyperthyroidism, Grave's disease, thyroid volume, goiter...
- (4) Based on eQTL analysis and summary-based Mendelian randomization the authors selected SLC17A4 and AADAT genes and confirmed their role with thyroid hormones signaling in vitro.

This is the largest meta-analysis of GWAS studies on TSH and FT4 levels and the results are of clear interest. However, the paper is very dense and the authors should consider reducing the results part in order to make it clearer. For instance, I think that the parts (1) and (2) of the secondary analyses is not necessary as there is no follow-up analyses of this and these results are not discussed. Also

analysis of the association between GRS and hypothyroidism/hyperthyroidism may be grouped with analysis between GRS and other traits.

Here are some points that could help the authors to make the manuscript clearer:

- In table S1, we can see that exclusion criteria differ between studies. For instance, some studies did not exclude subjects that under hormone replacement therapy. Could this influence the results?
- Similarly, some study has a familial design. Did the authors conducted sensitivity analyses with and without these familial studies?
- In table 1, it would be useful to add the number of subjects included in the analyses. Also I2 need to be defined and the p-value of the heterogeneity between studies should be added.
- It seems that there is some confusion between the words "locus" and "genes" in the whole paper. For instance, in table 1, the column "locus" refers probably to the closest gene as some variants are not located in a gene. In this column, we expect to find locus name such as: 3p20, 2q35, etc.
- It may be useful to add some forest plots indicating the effect of the variants by study and a measure of the heterogeneity. As the protocol of each study is different, we expect to find some heterogeneity between studies.
- Table S10: this table is a bit confusing as the list of SNPs is not completely the same than in Table 1. The authors should specify that table 1 only includes novel GWAS SNPs that replicated + additional independent hits in susceptibility loci.

In table S10, rs17398234 was flagged as "not replicated" but we can see that the discovery and the replication phases showed concordant effect direction. In line 604, the authors consider that a variant replicated if it shows concordant effect direction in both phases. This should be clarified.

Also proxy SNP in replication phase and its r2 with the SNP of interest need to be indicated

- In line 595, it is indicated that 23 SNPs associated to TSH and 20 SNPs associated to FT4 levels are shown in table S10, however there are respectively n=22 and n=19 in the table.
- Line 250: To make the results clearer, the authors should indicate here and also in the discussion if all previous GWAS SNPs were replicated in the meta-analysis and if there any overlapping samples between previously reported GWAS and this study.
- The different studies used for the discovery phase has imputed different numbers of SNPs and indicate different criteria of SNPs exclusion after imputation. The authors should indicate the number of SNPs that were analyzed in the meta-analysis of GWAS and how this subset of SNPs was defined.

- In the GWAS for hypothyroidism and hyperthyroidism, it may be useful to have a description of the distribution per age group and by sex for cases and controls.
- Table S7 and line 328: it is not clear on which data the analyses were performed while the authors investigated the association between TSH loci and autoimmune diseases.

Reviewer #3 (Remarks to the Author):

This manuscript describes large meta-analysis of genome-wide association studies (GWAS), seeking to identify variants in genetic loci associated with thyroid status (TSH, FT4 levels) and thyroid dysfunction. The authors identify novel genetic loci associated with TSH or FT4; they have tested variants in candidate genes identified in transgenic mice with these datasets; a separate GWAS for hypothyroidism (increased TSH) and hyperthyroidism (low TSH) and computation of a genetic risk score for hypothyroidism and hyperthyroidism; they have correlated variants in genetic loci with levels of expression of genes in different tissues; finally, the authors have functionally characterised products of two genes (SLC17A4, AADAT) showing that the former mediates cellular T4/T3 transport and that the latter can metabolise T4/T3. Overall, this is an important study which makes a significant contribution to the field, identifying several new genetic loci that could affect the thyroid axis, potentially providing new insights into thyroid physiology. I have the following specific comments:

1. I think it would be helpful for hyperthyroidism to be defined more precisely. Specifically, the authors could specify whether decreased TSH signifies subnormal levels or levels that were undetectable.
2. I think Figure 2 is difficult for the non-expert readership to understand. They should consider modifying the results and/or figure legend to improve this.
3. It would be useful to know how SLC17A4 compares with other known thyroid hormone transporters (e.g., MCT8 or MCT10) with regard to efficacy of thyroid hormone uptake.

4. It is not clear what the pyruvic acid metabolites (TK3, TK4) of thyroid hormone are? This should be specified.

Reviewer #4 (Remarks to the Author):

Teumer et al describe a genome-wide analyses of genes important in thyroid hormone regulation, performing an association study that includes 72,167 individuals. Genetic associations were found with 109 genes. Additionally, a novel thyroid hormone transporter, SLC17A4, and a metabolizing enzyme, AADAT, were identified and characterized in a preliminary fashion. The scope of the study is large and permitted identification of these novel associations as well as identifying new genes. The findings are novel and potentially important in understanding the regulation of normal thyroid function and thyroid disease. There are several areas of the data presentation and characterization that should be addressed to improve clarity and strengthen the conclusions.

1. The characterization of the findings as assessing thyroid function and dysfunction are potentially misleading. The study assessed genetic determinants of measurements of TSH and free T4 in individuals with normal thyroid function, and then examined determinants of individuals with thyroid disease, which are mostly all due to autoimmune disease. Since these are acquired diseases, the genetic associations are not with thyroid dysfunction, other than the underlying autoimmune disease. There are germ line genetic causes of thyroid hormone signaling defects, but these are extremely rare and not addressed by this study. The authors state, line 294, that the genetic variants may differ from variants that underly thyroid dysfunction (hypothyroidism or hyperthyroidism), but they should state that they will definitely differ. A clearer distinction between genetic association with thyroid function and thyroid disease, should be made.

2. What is the iodine status of the individuals analyzed or, if not available, from the country of origin of the studies included.

3. Since multiple studies were combined, how were TSH and Free T4 measurements from various assays accounted for? Were values compared as standard deviations from the reference range? Comparison among TSH assays is especially challenging.

4. The main stratification was by those individuals with normal reference range TSH and Free T4, compared to those above or below this range. Was the thyroid autoantibody status for patients with a normal reference range TSH and Free T4 assessed?

5. The association of TSH and FT4 was examined, but was there a separate analysis to examine TSH “set point,” TSH relative to FT4?

6. Figure 1-The figure title implies the TSH/FT4 ratio, but presumably the association is TSH and FT4 independently, this should be clarified.

7. Figure 2-Is the analysis of the GSR in patients with an elevated TSH (hypothyroidism) and suppressed TSH (hyperthyroidism)? Were all patients with a low or elevated TSH included regardless of etiology? What fraction of individuals in these categories had thyroid autoantibody measurements?

8. Figure 2 seems to show that the association of GSR with Free T4 in hypo and hyperthyroidism was very weak.

9. Figure 4- What is the tissue-type expression profile of SLS17A4? How does the activity compare with that of Mct8? It would be useful to include Mct8 as a control in the functional studies of SLS17A4.

10. Figure 5-The figures should be labeled with the product being produced/measured in each panel. The tissue-type expression profile of ADDAT is reported in the discussion as intestine, liver, and kidney. Are the pyruvic acid metabolites of T4 and T3 detectable in the bile or serum of humans or rodents? What is the relevance of this metabolite?

11. Lines 498-502-Given that we already know the major players in thyroid hormone transport and metabolism, it is a stretch to speculate that these novel, but apparently minor factors, will translate to novel therapies. This section should be deleted.

We thank the reviewers for their overall positive evaluation and the helpful comments and corrections of our manuscript. Please find below our answers to their comments.

Reviewer #1 (Remarks to the Author):

This is an interesting study arising from a large collaboration of established experts in the field. Given thyroid disease is common and the consequences of thyroid disease can influence key health outcomes including ischaemic heart disease and osteoporosis, this is of interest to individuals outside of thyroidology. This paper builds on earlier work by Porcu et al. and Taylor et al. and provides a substantial increase in SNPs identified and the improvements to our knowledge of the genetic architecture of thyroid function. The manuscript is well written and analyses are appropriate and the paper is likely to be widely cited. I have no major comments.

Introduction

Well written only a very minor point in that variation in thyroid even with the population reference range is associated with adverse outcomes, not just subclinical overt disease.

We agree that this should be included, which has now been done accordingly (p.7, l.235).

Discussion

A few points to consider to enhance the manuscript for readability

Given the extra variation in TSH explained, would it be potentially possible to identify to an extent what an individual's target TSH on levothyroxine is most likely to be based on their, could this be discussed in more detail ?

Given this increase in number of TSH and FT4 associated loci, we agree that future studies should start exploring the use of these markers to predict the individual HPT axis setpoint. This predicted setpoint could be used to guide treatment of thyroid diseases. This is important as despite normalized TSH and FT4 levels, a substantial part of treated patients still have persistent hypo- or hyperthyroid complaints, leading to a lower quality of life. This has now been discussed in the discussion section (p.19, l.508).

Whilst a substantial increase has been explained in the genetic architecture of thyroid function, there is still some "dark matter" would the authors speculate on where to find this, do we need even larger GWAS or should we be focussing on rare variants (MAF <1%)

We thank the Reviewer for raising this important point, with which we fully agree. We think that both options represent possibilities to address the "dark matter". Therefore, we added the search for rare SNPs and structural variants as well as for common SNPs in a larger sample as future projects to the end of the Discussion. Our ThyroidOmics Consortium provides a well-established infrastructure to address these research tasks (p.22, l.570).

Given the size of the study where the authors surprised that variants in THRB were not associated with changes in thyroid status were they surprised not to see any other genes

TSH and FT4 analyses were restricted to SNPs with a minor allele frequency > 1% to prevent spurious results given the current study design (i.e. selection of array type and imputation panel in

combination with the total sample size). We did not detect any associations with variants in *THRB* as the mutations described in *THRB* leading to thyroid hormone resistance have a frequency far below 1% (Dumitrescu & Refetoff, endotext.org, 2015), while the SNPs with a frequency >1% have not been consistently associated with thyroid function in previous studies (Medici et al., *Endocrine reviews* 2015). There were no other known thyroid hormone pathway genes, which have previously been consistently associated with thyroid function, that were not replicated in the current GWAS (Medici et al., *Endocrine reviews* 2015).

Reviewer #2 (Remarks to the Author):

Teumer et al. conducted a meta-analysis of GWAS on TSH and FT4 levels. In the discovery stage that included 54,288 individuals from 22 cohort studies, they reported 22 SNPs significantly associated to TSH levels and 19 SNPs associated to FT4 levels that were not reported in previous studies. Of these, they replicated 19 SNPs associated to TSH and 16 SNPs associated to FT4 levels in 10 replication cohorts (5 with imputed data n=9053 and 5 with genotyped data n=13330).

Then they conducted several secondary analyses:

(1) They selected candidate gene from a mouse model (26 genes), associated to mice abnormal physiology or morphology (51 genes). They found that variants from 4 of these genes were associated to TSH and FT4 levels in the discovery phase with p-values at the limit of the significance and 2 of these variants in genes CGA and TPO were replicated using the replication cohorts.

(2) Using the same data, they defined hypothyroidism and hyperthyroidism using TSH levels, and then they conducted a GWAS study for these traits using logistic regression. They identified 7 susceptibility loci for hypothyroidism and 8 susceptibility loci for hyperthyroidism. Some of these loci were reported previously, but not all.

(3) They defined genetic risk scores (GRS) for TSH levels and for FT4 levels and analyze the association with other phenotypes. They reported association with several traits including hypothyroidism, hyperthyroidism, Grave's disease, thyroid volume, goiter...

(4) Based on eQTL analysis and summary-based Mendelian randomization the authors selected SLC17A4 and AADAT genes and confirmed their role with thyroid hormones signaling in vitro.

This is the largest meta-analysis of GWAS studies on TSH and FT4 levels and the results are of clear interest. However, the paper is very dense and the authors should consider reducing the results part in order to make it clearer. For instance, I think that the parts (1) and (2) of the secondary analyses is not necessary as there is no follow-up analyses of this and these results are not discussed. Also analysis of the association between GRS and hypothyroidism/hyperthyroidism may be grouped with analysis between GRS and other traits.

The mouse model studies approach identified an additional novel TSH associated gene (CGA), and we therefore believe these results are worthwhile including in the manuscript. As we agree that the results section is dense, we summarized the mouse model studies in a few lines and moved the rest to the supplementary data. Additionally, in the updated discussion section, we pointed out that one

of the previous TSH associated genes (*TPO*) was significantly replicated in our study only through the mouse candidate gene approach.

As limited GWAS have been performed on abnormal thyroid function tests, we kept the increased and decreased TSH analyses in the main text. These results illustrate that variants with effects on thyroid function within the normal range can also extend beyond the normal range. In concordance with the reviewer's suggestion, we moved the figure on GRS and the risk of hypothyroidism/hyperthyroidism, as well as the GRS association analyses on hypo-/hyperthyroidism and Graves' disease to the general GRS part.

Here are some points that could help the authors to make the manuscript clearer:

- In table S1, we can see that exclusion criteria differ between studies. For instance, some studies did not exclude subjects that under hormone replacement therapy. Could this influence the results?

We thank the Reviewer for pointing out this issue. As defined in our analysis plan that was distributed to the cohorts, we requested (together with the other described exclusion criteria) to exclude individuals under thyroid hormone replacement therapy (defined as ATC code H03). However, a few cohorts could not apply this exclusion because this information was not available in these cohorts. We realized that some entries regarding the exclusion criteria in Supplemental Table 1 were missing, which has now been corrected. Furthermore, we indicated in the footnote that the large BBJ (Biobank Japan) study was used for Graves' disease lookup only and not for GWAS analyses.

Furthermore, in our analysis we assessed genetically modified thyroid hormone levels. Medication intake is not expected to be associated with the genotypes, therefore the thyroid hormone levels under medication might add random noise to the outcome, which would attenuate the association effects, but would not lead to false positive associations. However, no marked changes on the results are expected given the relatively low number of individuals under medication included in the GWAS due to missing information on medication within a minority of the studied cohorts. This was additionally supported as none of the associated SNPs showed a significant p-value for heterogeneity between the studies (after accounting for multiple testing).

- Similarly, some study has a familial design. Did the authors conducted sensitivity analyses with and without these familial studies?

The family structure of the respective cohorts was already taken into account during the association analyses, i.e. by inclusion of the kinship matrix as a random effect as described in the Methods section (GWAS in individual studies). This will correct the statistical analyses for possible family specific effects, and is in accordance to other GWAS projects. Therefore, we did not perform additional sensitivity analyses by excluding family studies. Furthermore, the interpretation of such sensitivity analyses would be difficult as it is accompanied by a substantial loss of statistical power.

- In table 1, it would be useful to add the number of subjects included in the analyses. Also I² need to be defined and the p-value of the heterogeneity between studies should be added.

We added the number of subjects and the p-value of the heterogeneity, as well as the definition of I² to Table 1.

- It seems that there is some confusion between the words “locus” and “genes” in the whole paper. For instance, in table 1, the column “locus” refers probably to the closest gene as some variants are not located in a gene. In this column, we expect to find locus name such as: 3p20, 2q35, etc.

We agree that this should be clarified. To clarify that the name of the locus refers to the closest gene throughout the paper (i.e. to be more specific compared to e.g. 3p20), we added the following sentence to the locus definition part of the methods (p.26, l.666):

“The loci were named according to the nearest gene of the index SNP.”

To be consistent in this way with the rest of the manuscript, we decided to keep the “locus” also in Table 1.

- It may be useful to add some forest plots indicating the effect of the variants by study and a measure of the heterogeneity. As the protocol of each study is different, we expect to find some heterogeneity between studies.

To provide information on heterogeneity, we added the I^2 values to the tables (where applicable) which provides a suitable measure for quantifying heterogeneity (Higgins *et al.*, BMJ (2003)). Additionally, we added to the table descriptions now, that the I^2 provides the percentage of total variation across studies that is due to heterogeneity. Finally, we provided forest plots of the novel associations in the new Supplementary Figures 3 and 4. None of these associated SNPs showed a significant p-value for heterogeneity after accounting for multiple testing.

- Table S10: this table is a bit confusing as the list of SNPs is not completely the same than in Table 1. The authors should specify that table 1 only includes novel GWAS SNPs that replicated + additional independent hits in susceptibility loci.

We agree and added the following clarifying sentence in accordance to the Reviewer’s suggestion to the beginning of the description of Table 1:

“The table contains the list of the index SNPs and additional independent associations of replicated susceptibility loci.”

In table S10, rs17398234 was flagged as “not replicated” but we can see that the discovery and the replication phases showed concordant effect direction. In line 604, the authors consider that a variant replicated if it shows concordant effect direction in both phases. This should be clarified.

As correctly pointed out by the Reviewer, the effect direction was concordant but the combined discovery+replication p-value was not genome-wide significant ($p < 5E-8$). To clarify this issue, we changed the table footnote to

“(*) did not replicate because not all criteria specified in the methods were fulfilled.”

whereas in the Methods we state:

“The results from the discovery meta-analysis and the results of replication studies were meta-analyzed to obtain the overall p-values of the selected SNPs. SNPs with p-values below genome-wide significance in this combined analysis and with concordant effect directions in both stages were considered as replicated.”

("and" was added to the last sentence for clarification).

Also proxy SNP in replication phase and its r2 with the SNP of interest need to be indicated

We added the requested information to Table S10 and indicated in the footnote, that only two cohorts (Health2006 and Inter99) used proxy SNPs. Additionally, we corrected a typo in the combined sample size provided for SNP rs56069042 (*MCR4*).

- In line 595, it is indicated that 23 SNPs associated to TSH and 20 SNPs associated to FT4 levels are shown in table S10, however there are respectively n=22 and n=19 in the table.

We thank the Reviewer for thoroughly checking these numbers and corrected the main text accordingly. Indeed, we originally selected the numbers for replication indicated in the text, but two of these SNPs were meanwhile discovered by Taylor *et al.* which was accidentally not updated in our Methods.

- Line 250: To make the results clearer, the authors should indicate here and also in the discussion if all previous GWAS SNPs were replicated in the meta-analysis and if there any overlapping samples between previously reported GWAS and this study.

Almost all of the previous GWAS associations replicated. To emphasize the robustness of the previous associations and to address the overlap of cohorts with our study, we added an extra paragraph to the Discussion section (p.17, l.461).

- The different studies used for the discovery phase has imputed different numbers of SNPs and indicate different criteria of SNPs exclusion after imputation. The authors should indicate the number of SNPs that were analyzed in the meta-analysis of GWAS and how this subset of SNPs was defined.

As described in the Methods section (Statistical methods for meta-analysis), the individual study GWAS result files were meta-analyzed by filtering SNPs with minor allele frequency ≤ 0.005 or an imputation quality score ≤ 0.4 before meta-analysis (i.e. by Metal during the meta-analysis computation). We added to the revised manuscript, that 9,653,808 SNPs per median passed this filter in each cohort (IQR: 9,302,604-10,705,092). Furthermore, the meta-analysis result SNPs that were present in less than 75% of the total sample size contributing to the respective meta-analysis (separately for autosomal and X-chromosomal SNPs) or with a (meta-analysis combined) MAF ≤ 0.01 were excluded from subsequent analyses. In the hypo- and hyperthyroidism GWAS, we increased this MAF to 0.05 because the low number of cases in the analysis caused spurious low-MAF findings. We specified the number of SNPs that passed these filters in detail now, and clarified the corresponding sentence:

"Finally, data for up to 8,048,941 genotyped or imputed autosomal and X-chromosomal SNPs were available after the discovery stage meta-analysis of TSH, FT4, and up to 5,965,951 SNPs after hypo- and hyperthyroidism."

NB: Because of filtering SNPs below 75% of the total sample size per GWAS, the final number of SNPs varies slightly depending on the trait and stratum analyzed.

- In the GWAS for hypothyroidism and hyperthyroidism, it may be useful to have a description of the distribution per age group and by sex for cases and controls.

We agree and added a new Supplementary Table 5 to provide the requested information:

hypothyroidism (% of all individuals)				hyperthyroidism (% of all individuals)			
0-19	male	0.0%	0.7%	0-19	male	0.0%	0.7%
0-19	female	0.0%	0.8%	0-19	female	0.0%	0.8%
20-39	male	0.2%	6.9%	20-39	male	0.2%	7.1%
20-39	female	0.4%	7.7%	20-39	female	0.3%	8.0%
40-59	male	0.6%	17.5%	40-59	male	0.6%	18.0%
40-59	female	1.3%	17.7%	40-59	female	0.7%	18.2%
60-79	male	1.0%	20.0%	60-79	male	0.7%	20.6%
60-79	female	2.1%	18.6%	60-79	female	0.8%	19.2%
80+	male	0.2%	1.5%	80+	male	0.1%	1.6%
80+	female	0.3%	2.3%	80+	female	0.1%	2.4%
0-19	male	0.0%	0.7%	0-19	male	0.0%	0.7%

The % values correspond to the total number of samples in the hypo- and hyperthyroidism analysis, respectively.

- Table S7 and line 328: it is not clear on which data the analyses were performed while the authors investigated the association between TSH loci and autoimmune diseases.

We added the relevant information to the former line 328, i.e. that the lookup for association with TPOab was performed in a former GWAS (Medici et al. PLoS Genetics 2014), and for Graves' disease in the BioBank Japan Project. Details of these studies are provided in the Methods section *Included studies*.

Reviewer #3 (Remarks to the Author):

This manuscript describes large meta-analysis of genome-wide association studies (GWAS), seeking to identify variants in genetic loci associated with thyroid status (TSH, FT4 levels) and thyroid dysfunction. The authors identify novel genetic loci associated with TSH or FT4; they have tested variants in candidate genes identified in transgenic mice with these datasets; a separate GWAS for hypothyroidism (increased TSH) and hyperthyroidism (low TSH) and computation of a genetic risk score for hypothyroidism and hyperthyroidism; they have correlated variants in genetic loci with levels of expression of genes in different tissues; finally, the authors have functionally characterised products of two genes (SLC17A4, AADAT) showing that the former mediates cellular T4/T3 transport and that the latter can metabolise T4/T3. Overall, this is an important study which makes a significant contribution to the field, identifying several new genetic loci that could affect the thyroid axis, potentially providing new insights into thyroid physiology. I have the following specific comments:

1. I think it would be helpful for hyperthyroidism to be defined more precisely. Specifically, the authors could specify whether decreased TSH signifies subnormal levels or levels that were undetectable.

In these analyses, cases were defined as having a TSH level below the assay reference range (p.10, l.297). Therefore this group not only includes subjects with fully suppressed TSH levels but also

subjects with more mild and subclinical forms of hyperthyroidism. The same holds true for the increased TSH GWAS. This has now been emphasized (p.10, l.300).

2. I think Figure 2 is difficult for the non-expert readership to understand. They should consider modifying the results and/or figure legend to improve this.

We agree. This has now been clarified in the figure legend.

3. It would be useful to know how SLC17A4 compares with other known thyroid hormone transporters (e.g., MCT8 or MCT10) with regard to efficacy of thyroid hormone uptake.

We agree with the Reviewer's comment and have now put more emphasis to the comparison of SLC17A4 with other currently known thyroid hormone transporters, in particular MCT8 (the most specific thyroid hormone transporter identified to date). Therefore, we have now included a direct comparison between the T3 and T4 transport by SLC17A4 and MCT8 in the absence and presence of CRYM and the results of saturation experiments with MCT8 as a supplemental figure 5. This will help the readership to put the results of SLC17A4 transport into perspective.

4. It is not clear what the pyruvic acid metabolites (TK3, TK4) of thyroid hormone are? This should be specified.

We have now included background information on the metabolism of thyroid hormones and more explicitly introduced the concept of alanine side-chain metabolism in the result and discussion section and included several relevant references on this topic. Although the existence of metabolic routes other than mono-deiodination has already been acknowledged in the 1950's, research on these pathways have been very limited ever since. Roche and colleagues already identified pyruvic acid metabolites in urine and bile of rats injected with radio-labelled T3 and T4, but the physiological relevance of this metabolic route and the responsible enzyme(s) have been unknown. Our studies show that AADAT is able to catalyze the conversion of T3 and T4 to TK3 and TK4, respectively. The finding that AADAT was among the hits of our GWAS study strongly suggests that this metabolic step is relevant in controlling serum (free) T4 concentrations in humans. This has now been added to the discussion section. (p.20, l.532)

Reviewer #4 (Remarks to the Author):

Teumer et al describe a genome-wide analyses of genes important in thyroid hormone regulation, performing an association study that includes 72,167 individuals. Genetic associations were found with 109 genes. Additionally, a novel thyroid hormone transporter, SLC17A4, and a metabolizing enzyme, AADAT, were identified and characterized in a preliminary fashion. The scope of the study is large and permitted identification of these novel associations as well as identifying new genes. The findings are novel and potentially important in understanding the regulation of normal thyroid function and thyroid disease. There are several areas of the data presentation and characterization that should be addressed to improve clarity and strengthen the conclusions.

1. The characterization of the findings as assessing thyroid function and dysfunction are potentially misleading. The study assessed genetic determinants of measurements of TSH and free T4 in individuals with normal thyroid function, and then examined determinants of individuals with thyroid

disease, which are mostly all due to autoimmune disease. Since these are acquired diseases, the genetic associations are not with thyroid dysfunction, other than the underlying autoimmune disease. There are germ line genetic causes of thyroid hormone signaling defects, but these are extremely rare and not addressed by this study. The authors state, line 294, that the genetic variants may differ from variants that underly thyroid dysfunction (hypothyroidism or hyperthyroidism), but they should state that they will definitely differ. A clearer distinction between genetic association with thyroid function and thyroid disease, should be made.

We agree with the Reviewer that this may be confusing. This is mainly due to the fact that the GWAS studies were performed in population-based cohorts, and not in cohorts of patients with a thyroid disease. Individuals on thyroid medication or a history of thyroid surgery were therefore excluded, resulting in a relative overrepresentation of individuals with subclinical forms of thyroid dysfunction. The identified variants are therefore expected to be a mix of variants which have been previously associated with hypo- or hyperthyroidism (e.g., *TPO*, *FOXE1* and *ATXN2*) and variants which lead to a TSH level which is slightly above or below the population-based reference ranges. These latter effects can either reflect true mild thyroid dysfunction with increased risk of clinical consequences or merely reflect a deviation from the individual thyroid setpoint with no clinical consequences. While our GRS analyses suggest that carrying multiple risk alleles leads to an increased risk of overt thyroid dysfunction and related clinical consequences, the exact contribution of each individual variant needs to be clarified in future studies. We re-phrased the statement in (former) line 294 to make a clearer distinction between thyroid function and disease.

2. What is the iodine status of the individuals analyzed or, if not available, from the country of origin of the studies included.

As data on iodine status were not available on an individual level, we added the iodine status of the studies' country of origin. For most studies, this was based on available WHO data (Supplementary Table 1).

3. Since multiple studies were combined, how were TSH and Free T4 measurements from various assays accounted for? Were values compared as standard deviations from the reference range? Comparison among TSH assays is especially challenging.

We undertook several steps to address the heterogeneity of the different assays. The sample exclusions that were based on the reference ranges were applied on a cohort specific level. This takes the assay specific values of each cohort into account, as well as the cohort specific reference values that were based on the cohort's epidemiological data (if available). Prior to the GWAS, the TSH and FT4 values of each cohort were inverse normal transformed (after applying the sample exclusion criteria), i.e. normalized to a standard deviation of 1 and aligned to a normal distribution. This makes the GWAS association results of the studies invariant to assay related shifts in the mean measures or their standard deviation and scale.

4. The main stratification was by those individuals with normal reference range TSH and Free T4, compared to those above or below this range. Was the thyroid autoantibody status for patients with a normal reference range TSH and Free T4 assessed?

Thyroid autoantibody status was not assessed in subjects with normal range TSH and FT4 levels. Detected associations with normal range TSH and/or FT4 levels might therefore also reflect mild pre-

forms of thyroid autoimmunity which have not yet led to thyroid dysfunction. We therefore tested the associated normal range TSH and FT4 hits with TPOab-positivity (p.11, l.324) and found only associations of variants in the *MAF*, *SPATA13* and *VAV3* genes. This has now been clarified in the manuscript (p.12, l.329).

5. The association of TSH and FT4 was examined, but was there a separate analysis to examine TSH “set point,” TSH relative to FT4?

Although it is expected that the variants with the largest effects on the HPT axis setpoint have also been detected in our separate normal range TSH and FT4 analyses, we agree that a dedicated setpoint analysis would be interesting. Unfortunately, we were limited by the fact that not all cohorts had available data on both TSH and FT4 levels. However, this analysis could be performed in future studies. This has now been mentioned in the discussion section, which nicely adds to the added section on HPT axis setpoint prediction (p.19, l.508).

6. Figure 1-The figure title implies the TSH/FT4 ratio, but presumably the association is TSH and FT4 independently, this should be clarified.

To avoid this misinterpretation, we changed the figure title to “TSH vs. FT4”.

7. Figure 2-Is the analysis of the GSR in patients with an elevated TSH (hypothyroidism) and suppressed TSH (hyperthyroidism)? Were all patients with a low or elevated TSH included regardless of etiology? What fraction of individuals in these categories had thyroid autoantibody measurements?

The former Figure 2 (now Figure 5) shows the GRS association analyses on cases with elevated TSH (hypothyroidism) and suppressed TSH (hyperthyroidism), respectively, whereas individuals having TSH values within the cohort specific reference range were used as controls (please see also our answer to question 1 of reviewer #3). We included all patients except individuals with thyroid medication (H03) or individuals that underwent thyroid surgery. Information on TPOab was assessed in approximately 50% of our studies.

8. Figures 2 seems to show that the association of GSR with Free T4 in hypo and hyperthyroidism was very weak.

There are indeed no significant associations between the FT4 GRS and the risk of increased or decreased TSH levels. This observation is in line with the fact that there is only partial overlap between the TSH and FT4 associated hits. For some FT4 associated hits, the lack of association with TSH levels can be explained on physiological grounds. E.g., the identified SNP in *DIO1* decreases the enzymatic activity of the protein, leading to less T4 to T3 conversion, resulting in higher T4 levels, but lower T3 levels. These opposite effects on T4 and T3 levels result in no net effect on feedback to the pituitary and therefore no effect on TSH levels. Similar hypotheses could be postulated for other loci involved in thyroid hormone metabolism, such as *DIO3OS* and *AADAT*, while the lack of effects on TSH levels of the other FT4 associated loci should be clarified in future studies. This has now been added (p.18, l.486).

9. Figure 4- What is the tissue-type expression profile of SLS17A4? How does the activity compare with that of Mct8? It would be useful to include Mct8 as a control in the functional studies of SLS17A4.

We agree with the Reviewer that a direct comparison to MCT8, the most specific thyroid hormone transporter identified to date, would be of great interest. Therefore, we have now included a direct comparison between the T3 and T4 transport by SLC17A4 and MCT8 in the absence and presence of CRYM and the results of saturation experiments with MCT8 as a Supplementary Figure 5. This will help the readership to put the results of SLC17A4 transport into perspective. In addition, we have now referred to studies of Togawa et al., who reported on the tissue distribution of SLC17A4 mRNA expression in human at the appropriate locations in the results and discussion section.

10. Figure 5-The figures should be labeled with the product being produced/measured in each panel. The tissue-type expression profile of ADDAT is reported in the discussion as intestine, liver, and kidney. Are the pyruvic acid metabolites of T4 and T3 detectable in the bile or serum of humans or rodents? What is the relevance of this metabolite?

We thank the Reviewer for this suggestion and now made reference to studies of Roche and colleagues performed in the 1950s in which they demonstrated TK3 and TK4 in urine and bile from rat injected with radio-labelled T3 and T4, respectively. Although the existence of metabolic routes other than mono-deiodination have already been acknowledged in the 1950's, research on these pathways have been very limited ever since. Our studies at least suggest that SNPs that potentially affect AADAT expression levels may alter serum free T4 concentrations, indicating that the transamination of thyroid hormone might be a relevant metabolic pathway in the control of serum thyroid hormone levels. The labels of Figure 5 have now been adapted as requested.

11. Lines 498-502-Given that we already know the major players in thyroid hormone transport and metabolism, it is a stretch to speculate that these novel, but apparently minor factors, will translate to novel therapies. This section should be deleted.

We agree that major players in thyroid hormone transport and metabolism have been identified, however parts remain incompletely understood (Refetoff, Endotext (2000) and Peeters & Visser, Endotext (2000)). Although the effects of the identified polymorphisms are small, the gene itself can still play an important role. In fact, our *in vitro* analyses show that *SLC17A4* is a potent thyroid hormone transporter, while *AADAT* is a potent thyroid hormone metabolizing enzyme. We therefore believe it is fair to state that our findings open up a potential new avenue in this field.

Reviewer #1 (Remarks to the Author):

Many thanks for addressing my comments. I am satisfied with all the responses. I have also reviewed the comments and responses to the other reviewers and I am happy with them also.

Reviewer #2 (Remarks to the Author):

The authors replied adequately to all my comments and the manuscript is much clearer.

I have no other comment to make.

Reviewer #3 (Remarks to the Author):

The authors have clarified definitions of thyroid dysfunction and better explained computation of risk scores.

They have compared function of SLC17A4 with a known thyroid hormone transporter (MCT8).

They have provided more background about pyruvate metabolites of thyroid hormone.

Reviewer #4 (Remarks to the Author):

The authors have responded to the reviewer comments and made appropriate additions and modifications in the revised manuscript.

Point-by-point response to the requested format changes on the manuscript NCOMMS-18-05428A

* Nature Communications uses a transparent peer review system, where for manuscripts submitted from January 2016 we are publishing the reviewer comments to the authors and author rebuttal letters of our research articles online as a supplementary peer review file. Please let us know in the cover letter when submitting the final version of your manuscript if you wish to opt out of this scheme or not. If you are concerned about the release of confidential data, we do permit redactions in the interest of confidentiality. If you would like to remove such information from these files, then please let us know specifically what information you would like to have removed. Please note that we cannot incorporate redactions for other reasons. Reviewer names will be published in the peer review files if the reviewer comments to the authors are signed by the reviewer, or if reviewers explicitly agree to release their name. For more information, please refer to our FAQ page at:

<https://media.nature.com/full/nature-assets/ncomms/authors/ncomms-transparent-peer-review.pdf>

As indicated in the cover letter we have no objections, and confirm that the comments include no confidential information that needs to be removed.

* Please ensure that an updated editorial policy checklist that verifies compliance with all required editorial policies is completed and uploaded with the revised article. All points on the policy checklist must be addressed; if needed, please revise your manuscript in response to these points. Please note that this form is a dynamic 'smart pdf' and must therefore be downloaded and completed in Adobe Reader.

Editorial policy checklist: <https://www.nature.com/authors/policies/Policy.pdf>

Done and uploaded.

* Your manuscript should comply with our policies and format requirements, detailed in our checklist for authors at:

http://www.nature.com/article-assets/npg/ncomms/authors/ncomms_manuscript_checklist.pdf

Done and uploaded.

* Data availability statements and data citations policy: All Nature Communications manuscripts must include a section titled "Data Availability" as a separate section after the Methods section but before the References. For more information on this policy, and a list of examples, please see <http://www.nature.com/authors/policies/data/data-availability-statements-data-citations.pdf>

- Accession codes for deposited data
- Other unique identifiers (such as DOIs and hyperlinks for any other datasets)

- At a minimum, a statement confirming that all relevant data are available from the authors
- If applicable, a statement regarding data available with restrictions
- If a dataset has a Digital Object Identifier (DOI) as its unique identifier, we strongly encourage including this in the Reference list and citing the dataset in the Data Availability Statement.

The GWAMA summary results data are currently in the process of being uploaded to the CHARGE dbGaP website under accession phs000930 [<https://www.ncbi.nlm.nih.gov/gap>] and in the locuszoom web page [<http://locuszoom.sph.umich.edu/genform.php>]. We added the corresponding information to the manuscript in the Data availability section.

* DATA SOURCES: We strongly encourage authors to deposit all new data associated with the paper in a persistent repository where they can be freely and enduringly accessed. We recommend submitting the data to discipline-specific, community-recognized repositories, where possible and a list of recommended repositories is provided here: <http://www.nature.com/sdata/policies/repositories>

If a community resource is unavailable, data can be submitted to generalist repositories such as figshare (<https://figshare.com/>) or Dryad Digital Repository (<http://datadryad.org/>). Please provide a unique identifier for the data (for example a DOI or a permanent URL) in the data availability statement, if possible. If the repository does not provide identifiers, we encourage authors to supply the search terms that will return the data. For data that have been obtained from publically available sources, please provide a URL and the specific data product name in the data availability statement. Data with a DOI should be further cited in the methods reference section.

Please refer to our data policies here: <http://www.nature.com/authors/policies/availability.html>

We will upload the GWAMA summary results files to CHARGE dbGaP website under accession phs000930.

* To ensure correct hyperlinking of the accession codes in your manuscript, please add the hyperlink or DOI in square brackets directly after the code throughout (for example, '5XRN [<http://dx.doi.org/10.2210/pdb5XRN/pdb>]', '1483958 [<https://dx.doi.org/10.5517/ccdc.csd.cc1lt5m6>]', 'SRP109982 [<https://www.ncbi.nlm.nih.gov/sra/?term=SRP109982>]' or 'NQLW00000000 [https://www.ncbi.nlm.nih.gov/assembly/GCA_002312845.1/]').

Done.

* Please check whether your manuscript or Supplementary Information contain third-party images, such as figures from the literature, stock photos, clip art or commercial satellite and map data. We strongly discourage the use or adaptation of previously published images, but if this is unavoidable, please request the necessary rights documentation to re-use such material from the relevant copyright holders and return this to us when you submit your revised manuscript.

In particular, please indicate whether you or a co-author created the beakers in Supp. Fig. 1

To rule out any potential copyright violation, our co-author Yong Li has drawn now own beaker images which replaced the former ones in the revised version of the manuscript.

* Nature journals require authors of life sciences research papers to include relevant details about several elements of experimental and analytical design in their manuscripts. This initiative aims to improve the transparency of reporting and the reproducibility of published results and is described at: <http://www.nature.com/authors/policies/reporting.pdf> To ensure that your manuscript complies with our policy, please complete our checklist for authors: <https://www.nature.com/authors/policies/ReportingSummary.pdf>

You may also find the following collection of articles on statistics for biologists helpful: <http://www.nature.com/collections/qghhqm>

Done and uploaded.

* If a consortium is included in the main author list, all members of the consortium are considered bona fide authors, and must be listed together with their affiliations at the end of the main Article (not in the Supplementary Information). However, if a member of the consortium already appears as an individual name in the main author list, then his/her name should not be listed again at the end of the Article. If you need to give credit to a consortium, a project or a group of people who do not meet authorship criteria, you can add a mention in the Acknowledgements section or elsewhere (in which case, a full list of members can be provided as a Supplementary Note in the Supplementary Information, if desired). For guidelines on authorship and consortia, please visit: <http://www.nature.com/authors/policies/authorship.html>

We applied the requested changes accordingly. This includes that the ThyroidOmics Consortium name was moved from author list to the acknowledgements, and that the LifeLines Cohort Study authors are listed now together with their affiliations at the end of the manuscript.

* For reasons of journal style and clarity, I would like to suggest a revision to the title. I would be happy to consider alternative suggestions; please ensure that the title does not exceed 15 words and does not contain punctuation.

"Genome-wide analyses identify a role for SLC17A4 and AADAT in thyroid hormone regulation"

We agree with the suggested title and changed it accordingly.

* Please shorten the abstract to 150 words or fewer. It should be accessible and include the background and context of the work, 'Here we report' or an equivalent phrase, and then the major results and conclusions of the paper written in the present tense. It must not contain references or unnecessary acronyms/abbreviations.

We changed the abstract accordingly and changed it to present tense.

* To comply with our Article templates, the text must be split into the following sections: Introduction and Results, with optional Discussion and Methods. The Introduction section must include the background and rationale for the work, and the final paragraph should be a brief summary of the major results and conclusions; the results of the current study should only be discussed in this final paragraph. The Results section must be split into subheaded sections, ensuring that the subheadings are no longer than 60 characters including spaces.

We added the requested section headings, and renamed the section "Identification of a novel thyroid hormone transporter and a metabolizing enzyme" to "In vitro studies" to not extend the

character limit. Additionally, we added a paragraph summarizing the major results and conclusions to the end of the Introduction.

* The Introduction should be structured in such a way that the background and rationale for the study, including all previous literature, is discussed first, and the current work is discussed only in the final paragraph. Please rearrange the Introduction to adhere to this format: all necessary background and context should be in preceding paragraphs and the major results and conclusions of the current work should be summarised in present tense in the final paragraph.

We applied the requested changes and added a final paragraph summarizing the major results and conclusions in present tense to the end of the Introduction.

* Please divide the Results and Methods section into subsections, each with a title of 60 characters or fewer including spaces. Please ensure that a subheading is present at the very beginning of both the Results and Methods sections, even if there is only one subsection.

We applied the requested changes.

* Please provide a full Methods section in the main manuscript file. Please note that there are no word limits to the Methods section. The Methods section should contain subheadings that contain fewer than 60 characters including spaces. Please bring all "Supplementary Methods" forward to the main article file.

We moved all methods from the Supplemental material to the main manuscript, and arranged their order as they appear in the Results section.

* Please remove phrases such as 'new', 'novel', 'for the first time', 'unprecedented', etc. as these are not needed to emphasise the importance of your work.

We checked the manuscript and removed several of these occurrences.

* In the Methods, please provide sufficient information such that the experiments could reasonably be reproduced without reference to other papers, and avoid use of the term 'as described previously'.

We ensured that the methods can be reproduced as described in the manuscript and limited the references to other papers for applied standard methods which are established already for several decades.

* We are committed to ensuring clarity and avoiding ambiguity in the mathematics in our papers. Consequently, please carefully check the mathematical terms throughout your manuscript and Supplementary Information (including labels on figures and figure captions) to ensure that it conforms strictly to the following guidelines. Equations should be supplied in editable format, and not as images. In mathematical terms, scalar variables (e.g. x , V , χ) should be typeset in italic, whereas multi-letter variables should be formatted without italic. Constants (e.g. \hbar , G , c) should be typeset in italics (the only exceptions being e , i , π , which should be typeset without italic) and vectors (such as r , the wavevector k , or the magnetic field vector B) should be typeset in bold without italics. In contrast, subscripts and superscripts should only be italicized if they too are variables or constants. Those that are labels (such as the 'c' in the critical temperature, T_c , the 'F' in the

Fermi energy, E_F , or the 'crit' in the critical current, I_{crit}) should be typeset in roman. Please also ensure the same convention is followed in figure labels, axes, and such. Additionally, to avoid doubt, unit dimensions should be expressed using negative integers (e.g. $\text{kg m}^{-1} \text{s}^{-2}$ not kg/ms^2) or the word 'per'.

We corrected the italic letters in the mathematical terms, and the negative unit dimensions accordingly.

* Please ensure that italics are used for scalar variables, physical constants, species and gene names, and bold font is used for numbering chemical compounds. Italics and bold font should not be used for emphasis.

We corrected the italic letters in the mathematical formulas accordingly.

* With regards to the experiments using human participants or data, please confirm that you have complied with all relevant ethical regulations and that a statement affirming this, and the name of the board and institution that approved the study protocol, is included in the methods section of the manuscript. Please also ensure that a statement is made in the methods confirming that informed consent was obtained from all human participants.

A corresponding statement is included in the first paragraph of the Methods section.

* Please ensure that the origins of all cell lines used are stated in the methods section (ATCC, vendor, etc).

This information is included as requested.

* If the work involves any cancer cell lines that are listed in the database of commonly misidentified cell lines, ICLAC (<http://iclac.org/databases/cross-contaminations>), please provide justification for their use in the methods section. Please also state where the lines were obtained from; whether they were tested for mycoplasma contamination; and whether they were authenticated, and if so, by which method.

Our study does not include cancer cell lines.

* Please ensure that +/- values are defined at the first point of use within the text and figure legends and numbers of replicates are given.

We added the corresponding sample size (n) to the manuscript and figure legends where missing.

* In each Figure and Supplementary Figure where error bars are used, they must be defined, and the number of experimental replicates stated. One statement at the end of each figure is sufficient if the error bars are equivalent throughout the figure.

The requested information is provided.

* Wherever p-values are stated in the main text and figure legends, please also state the name of the statistical test.

We added the statistical test to the figure legends where missing (e.g. to the Manhattan and QQ plots).

* Where p-values are presented as symbols/letters, please ensure that all symbols/letters are defined in the relevant figure legend, together with the statistical test used.

Checked.

* Please ensure that all colour scales are defined in either the figure or its associated legend.

Checked.

* Please ensure that the corresponding dot plots are overlaid in the bar charts.

Checked.

* Please ensure that each display item is no larger than a single A4 portrait page (260x179 mm).

Checked.

* Please define any new abbreviations, symbols or colours present in your figures in the associated legends, noting that these should be written out in words (blue circles, red dashed line, etc.) as symbols will not appear properly in the HTML text.

Checked.

* In order for us to accurately represent the data in your tables, they must conform to journal style; unless you format the tables in your manuscript as described here, they won't display correctly in the published paper. Data in tables must be free from bold/italic formatting unless this has been clearly defined in the footnote. We are also unable to process colour in tables so this should be removed. We cannot display tables that do not fit onto a single page or multi-element tables. Finally, we are unable to merge cells or include vertical dividing lines or diagonal lines.

Checked. Table 1 contained accidentally red colored values (indicating changes in the last revision) which were set to black color now.

* Please ensure that Table 1 fits on a single A4 page in portrait orientation.

We re-formatted Table 1 accordingly to fit the requested size.

* Please ensure the references are in the standard Nature format and follow the sequence: author list, title of paper, name of journal, volume number, initial-final page numbers or article number (year). Please note that dois are required only for online-only publications and correct journal abbreviations should be given.

We checked the references, and removed two DOIs by updating the citation information.

* Please supply an acknowledgements section after the Methods section.

Done.

* The acknowledgements section should be brief and should not include thanks to Editors or referees, effusive comments or dedications. Please add an "Acknowledgements" section after the references. Please include here the sentence "This manuscript is dedicated to the memory of Prof.dr.ir. Theo J. Visser, a brilliant scientist and mentor, who recently passed away.", not after the

list of affiliations. Please cite here the extended acknowledgements in the Supplementary Information (e.g. "Extended acknowledgements and study-specific acknowledgements are provided in Supplementary Note x." – please make sure to number all Supplementary Notes in the order in which they appear)

We applied the changes accordingly, and changed in the acknowledgements in the main text: "Extended acknowledgements and study-specific acknowledgements are provided in the Supplementary Note 4."

* Please make a 'Competing Interests' statement after the 'Author Contributions' section that refers to all authors. If there are no competing interests, please add the statement "The authors declare no competing interests."

We applied the statement accordingly.

* Please note that we do not reformat Supplementary Information files; they will be uploaded with the published article as they are submitted with the final version of your manuscript. Please check everything very carefully and remove any track changes from the file. Failure to adhere to our style guidelines will result in delays in production. The only sections we permit in the Supplementary Information file are Supplementary Figures, Supplementary Tables, Supplementary Methods, Supplementary Notes, Supplementary Discussion, Supplementary References.

We applied the changes accordingly.

* In the Supplementary Information file, please ensure that supplementary items are labelled and cited using only the following formats: Supplementary Figure 1, Supplementary Table 1, Supplementary Methods, Supplementary Note 1, Supplementary Discussion, Supplementary References. Please note the use of 'Supplementary' and that we do not use the 'S' prefix.

We changed the labels as requested to Supplementary Figure XX, Supplementary Table XX, and Supplementary Note XX.

* Throughout the main manuscript text, please use the following formats to cite Supplementary items: Supplementary Figure 1, Supplementary Table 1, Supplementary Methods, Supplementary Note 1, Supplementary Discussion, Supplementary References, Supplementary Movie 1, Supplementary Audio 1, Supplementary Data 1, Supplementary Software 1. Please note the use of 'Supplementary' and that we do not use the 'S' prefix.

We changed the citations as requested to Supplementary Figure XX, Supplementary Table XX, Supplementary Note XX, and Supplementary Data XX.

* We only permit one Supplementary Information file; please merge all Supplementary Information files into one PDF document. Only Supplementary Data/Software/Movie/Audio files should be submitted separately from the Supplementary Information.

We merged all Supplementary Information to a single pdf, and to 6 additional Supplementary Data Excel files.

* Each Supplementary Figure should be accompanied by a legend, which **should be presented below the figure** and may be up to 350 words, that refers to all panels within the figure, and a title that

summarises the figure and does not refer to specific panels. This also applies to spectra, which should be labelled as Supplementary Figures.

We now moved the legend below each Supplementary Figure.

* Please ensure that the Supplementary References appear at the end of the SI, and are self-contained and numbered from 1. References mentioned in both the main text and the Supplementary Information should be part of both reference lists so that the Supplementary Information does not refer to the reference list in the main paper and vice versa.

Checked.

* Large datasets should be supplied as Supplementary Data files, whereas smaller tables should be supplied as Supplementary Tables.

As confirmed by personal correspondence, we provide the large former Supplementary Tables 1, 4, 8, 9, 15 and 16 as separate Supplementary Data Excel files on a single sheet. All remaining Supplementary Tables were included now in the single Supplementary pdf file.

* Please supply legends for each Supplementary Data file in your cover letter (not in the Supplementary Information file).

The description of the six files is provided at the end of the cover letter.

* Your paper will be accompanied by a two-sentence editor's summary, of between 250-300 characters, when it is published on our homepage. Could you please approve the draft summary below or provide us with a suitably edited version.

“Thyroid dysfunction is a common public health problem and associated with cardiovascular co-morbidities. Here, the authors carry out genome-wide meta-analysis for thyroid hormone (TH) levels, hyper- and hypothyroidism and identify SLC17A4 as a TH transporter and AADAT as a TH metabolizing enzyme.”

We approve the suggested summary, thank you.